# LolA and LolB are conserved in Bacteroidota and are crucial for gliding motility and Type IX secretion
Tom De Smet[1], Elisabeth Baland[1,5], Fabio Giovannercole[1], Julien Mignon[2], Laura Lizen[1,6], Rémy Dugauquier[1], Frédéric Lauber [1,7], Marc Dieu [3], Gipsi Lima-Mendez [1], Catherine Michaux [2], Damien Devos [4,8] & Francesco Renzi [1] ✉

Lipoproteins are key outer membrane (OM) components in Gram-negative bacteria, essential for functions like membrane biogenesis and virulence. Bacteroidota, a diverse and widespread phylum, produce numerous OM lipoproteins that play vital roles in nutrient acquisition, Type IX secretion system (T9SS), and gliding motility. In *Escherichia coli*, lipoprotein transport to the OM is mediated by the Lol system, where LolA shuttles lipoproteins to LolB, which anchors them in the OM. However, LolB homologs were previously thought to be limited to γ- and β-proteobacteria. This study uncovers the presence of LolB in Bacteroidota and demonstrates that multiple LolA and LolB proteins co-exist in various species. Specifically, in *Flavobacterium johnsoniae*, LolA1 and LolB1 transport gliding motility and T9SS lipoproteins to the OM. Notably, these proteins are not interchangeable with their *E. coli* counterparts, indicating functional specialization. Some lipoproteins still localize to the OM in the absence of LolA and LolB, suggesting the existence of alternative transport pathways in Bacteroidota. This points to a more complex lipoprotein transport system in Bacteroidota compared to other Gram-negative bacteria. These findings reveal previously unrecognized lipoprotein transport mechanisms in Bacteroidota and suggest that this phylum has evolved unique strategies to manage the essential task of lipoprotein localization.

The cell envelope of Gram-negative bacteria is composed of two membranes: the inner membrane (IM) delimiting the cytoplasm, and the outer membrane (OM) delimiting the periplasm. Residing in between these two membranes is an additional thin layer made of peptidoglycan. Unlike the IM, composed exclusively of phospholipids, the OM is an asymmetric bilayer, with phospholipids on the inner and lipopolysaccharide (LPS) on the outer leaflet, forming a dense network impermeable to most compounds[1]. Import of nutrients through the OM thus relies on integral outer membrane proteins called porins, water-filled channels that allow substrates to cross the OM. In addition, lipoproteins, proteins with an acylated N-terminus allowing their anchorage in the membrane, are found in both membranes and participate in many functions, including nutrient acquisition, stress sensing, cell morphology, transport, etc.[1]. In *E. coli*, the most abundant lipoprotein is Lpp, inserted into the inner leaflet of the OM, which anchors the peptidoglycan to the OM thus playing a crucial role in the envelope architecture[2]. OM lipoproteins are also part of membrane protein complexes crucial for envelope maturation and cell viability such as the BAM complex, responsible for β-barrel protein insertion and folding into the OM[3] and the Lpt complex involved in LPS transport across the OM[4]. While in most studied bacteria, OM lipoproteins are mainly anchored facing the periplasm, it has recently been shown that surface-exposed lipoproteins are more widespread than initially thought, especially in bacteria of the phylum Bacteroidota (formerly Bacteroidetes), where they are crucial for nutrient acquisition[5–7]. Furthermore, OM lipoproteins are key elements of

[1]Research Unit in Biology of Microorganisms (URBM), Namur Research Institute for Life Sciences (Narilis), University of Namur, Namur, Belgium. [2]Laboratoire de Chimie Physique des Biomolécules, UCPTS, Namur Institute of Structured Matter (NISM), Namur Research Institute for Life Sciences (NARILIS), University of Namur, Namur, Belgium. [3]Technological Platform Mass Spectrometry Service (MaSUN), Namur Research Institute for Life Sciences (Narilis), University of Namur, Namur, Belgium. [4]Centro Andaluz de Biología del Desarrollo (CSIC), Universidad Pablo de Olavide, Sevilla, Spain. [5]Present address: Department of Chemistry, Umeå University, Umeå, Sweden. [6]Present address: Laboratoire de Chimie Bactérienne (LCB) CNRS-Aix-Marseille University, Marseille, France. [7]Present address: De Duve Institute, UCLouvain, Brussels, Belgium. [8]Present address: Center for Infection and Immunity of Lille, Pasteur Institute, Lille, France. ✉e-mail: francesco.renzi@unamur.be

the gliding and Type 9 secretion systems (T9SS), two intertwined types of machinery that are a hallmark for and unique to Bacteroidota[8].

Despite different localizations, lipoproteins share a common maturation pathway. They are synthesized in the cytoplasm and after translocation to the periplasm, mainly through the Sec machinery, their N-terminal cysteine is acylated by the action of three IM enzymes. Once lipidated, lipoproteins remain either in the IM or are targeted to the OM[9]. The periplasm is a hydrophilic aqueous environment, yet, lipoproteins having a hydrophobic moiety, they require a specific transport system[1,10].

In *E. coli*, lipoproteins are transported to the OM by the Lol system, composed of five proteins: the LolCDE *ATP*-binding cassette (ABC) transporter, responsible for lipoprotein extraction from the IM; the periplasmic chaperone LolA, which transports lipoproteins through the periplasm and delivers them to the OM lipoprotein LolB that inserts them in the OM[11]. Strikingly, while the components of the Lol machinery are essential for cell viability in *E. coli*, they are only partially conserved throughout Gram-negative bacteria. Indeed, LolB seems to be present only in β- and γ-Proteobacteria[12], while a bifunctional LolA, capable of both chaperoning and inserting lipoproteins in the OM, has recently been identified in α-Proteobacteria and Spirochetes[13,14]. In addition, bacteria such as Spirochetes and Bacteroidota abundantly expose lipoproteins on their surface[5,9,15].

In Bacteroidota, lipoproteins are targeted to the surface by an N-terminal lipoprotein export signal (LES), however, how they cross the OM is still unknown[16,17]. In Spirochetes, such an export signal could not be identified, but lipoprotein surface localization is dependent on an Lpt-like pathway where a homolog of the LPS transporter LptD is responsible for lipoprotein surface translocation[15]. It remains, therefore, unknown how Bacteroidota insert lipoproteins into their OM and how they flip some of them to the surface.

Here, by in silico distant homolog prediction, we identify LolB homologs in bacteria of the phylum Bacteroidota and demonstrate that multiple LolA and LolB proteins co-exist within the same Bacteroidota species.

We show that in the Bacteroidota *Flavobacterium johnsoniae*, which encodes three LolA and two LolB homologs, one LolA and one LolB are involved in lipoprotein trafficking. Deletion of either of these determines a lack of gliding and T9 protein secretion and severe membrane alterations. Strikingly, despite the presence of several LolA and LolB homologs, we provide evidence that surface lipoprotein transport can work independently of these proteins in *F. johnsoniae*. Finally, the finding that *F. johnsoniae* can survive in the absence of all LolA and LolB proteins strongly supports the hypothesis that another pathway can transport OM lipoproteins in the absence of LolA and LolB in Bacteroidota.

## Results
### *Flavobacterium johnsoniae* encodes several LolA and LolB proteins

While the presence of LolA homologs has been reported in Bacteroidota, no LolB homologs have been identified in bacteria of this phylum by sequence similarity searches. We started from the hypothesis that Bacteroidota might encode distantly related LolB homologs with low sequence identity, but that would share a similar structure. To identify such homologs, we performed an in silico prediction analysis searching for remote homologs of *E. coli* LolB in *F. johnsoniae* (see Methods for more details).

This analysis identified two LolB homologs in *F. johnsoniae*: Fjoh_1066 (LolB1) and Fjoh_1084 (LolB2). Both candidates are predicted to encode an SPII signal sequence, in accordance with LolB being a lipoprotein. We performed the same analysis to search for *E. coli* LolA homologs and three proteins were identified: the two LolA already found by sequence similarity and reported in the literature (Fjoh_2111, LolA1, and Fjoh_1085, LolA2)[18,19] and a third one, Fjoh_0605, LolA3. LolA is a periplasmic carrier and thus harbors an SPI signal allowing it to cross the IM via the Sec pathway and reach the periplasm[20]. All three LolA homologs carry such a signal sequence and are thus likely localized in the periplasm.

While genes *Fjoh_2111* (*lolA1*), *Fjoh_0605* (*lolA3*), and *Fjoh_1066* (*lolB1*) are encoded in different genomic loci, genes *Fjoh_1084* (*lolB2*) and

*Fjoh_1085* (*lolA2*) are part of an operon responsible for flexirubin biosynthesis. Flexirubin is thought to be localized in the OM, and thus LolA2 and LolB2 might be involved in the synthesis and/or transport of this pigment[19,21].

Next, we modeled the *F. johnsoniae* LolA and LolB protein structures and compared their overall secondary and tertiary structures with respect to the crystal structures of their *E. coli* homologs (Fig. 1). *F. johnsoniae* LolA and LolB variants retain the main features of the *E. coli* unclosed β-barrel presenting a convex and a concave side, composed of 11 antiparallel β-strands in which helices (α- and/or $3_{10}$-helices) are embedded (Fig. 1), and enclosed by an N-terminal α-helix. Among the LolA proteins, *E. coli* LolA (LolA$_{Ec}$) and LolA1 have the same mainly disordered C-terminal region with one short $3_{10}$-helix as well as one extra parallel β-strand, while LolA2 and LolA3 have a shorter and longer C-terminal end, respectively. In addition, the *E. coli* LolB (LolB$_{Ec}$) specific feature, consisting of a protruding loop comprising an exposed hydrophobic amino acid, is conserved in LolB1 and LolB2 (Fig. 1). In LolB$_{Ec}$ and LolB1 this is a leucine, while in LolB2 a phenylalanine. LolB2 is structurally closer to LolB$_{Ec}$ than LolB1 which has a long N- and C-terminal disordered tail, as well as a very long loop extension compacted beneath the barrel.

The mechanism of transfer of lipoproteins from LolA$_{Ec}$ to LolB$_{Ec}$ occurs via a complex formation through electrostatic potential complementarity, forming a contiguous hydrophobic surface, i.e., a tunnel-like structure, composed of both LolA and LolB concave sides. The lipoprotein is transferred due to a higher affinity of its lipid moiety for LolB than LolA due to a hydrophobic gradient between the cavities of the two partners[22]. LolA$_{Ec}$ concave side, especially along the edge of the β-barrel opening, is enriched in negatively charged amino acids, interacting with the LolB$_{Ec}$ convex side, which is mainly positively charged[22]. With this mechanism in mind, we compared the hydrophobicity (Supplementary Fig 1) and charge state (Supplementary Fig 2) of *F. johnsoniae* LolA and LolB homologs to highlight differences and, potentially predict LolA–LolB interactions.

An interesting tendency is a distinct enrichment in highly hydrophobic (leucine and isoleucine) amino acids highlighted for all the *F. johnsoniae* LolA and LolB homologs (Supplementary Table 1). Consequently, these are overall more hydrophobic than LolA$_{Ec}$ and LolB$_{Ec}$, and LolA2 stands out as the most hydrophobic one. Furthermore, LolB1 contains a higher proportion of hydrophobic residues than LolA1, suggesting a favorable hydrophobic gradient for the lipoprotein transfer between LolA1 and LolB1. Although LolB2 has fewer hydrophobic residues than LolA2, its higher content in leucine and their local concentration within the concave side of the β-barrel could nonetheless allow the lipoprotein transfer toward LolB2 (Supplementary Fig 1). The same rationale can be applied to the putative LolA1-LolB1 couple. In the case of LolA3, the side chains pointing toward the inside of the concavity are substituted in tyrosine and isoleucine, increasing the overall polarity of the binding site. Regarding the electrostatic-driven LolAB paired complexes formation, a slight decrease in arginine compensated by a large increase in the lysine content is observed for all the *F. johnsoniae* proteins (Supplementary Table 1), especially for LolA1 and LolA2, changing the concave side charge from negative to mostly positive (Supplementary Fig 2). LolA3 seems closer to LolA$_{Ec}$, with a predominance of aspartate and glutamate residues on the concave side, giving it a global negative charge. Most remarkably, its unique long and disordered C-terminal region bears a high positive charge, due to a dense population of lysine. Furthermore, while the positive character of the LolB convex side is mainly conserved in its *F. johnsoniae* homologs, a more evocative negative charge is exhibited by the concave side, particularly at the level of the loop extension of LolB1, absent in LolB$_{Ec}$ and LolB2 (Supplementary Fig 2). Therefore, the mechanism and preferential orientation by which LolA-LolB complexes are formed in *F. johnsoniae* most likely differ from that of *E. coli*, and pair selectivity will also arise from subtle modifications in hydrophobicity and charge distribution.

### Deletion of *lolA1* and *lolB1* affects gliding motility and Type IX secretion

Transposon mutagenesis screens to identify gliding motility genes of *F. johnsoniae* and of another member of the Bacteroidota resulted in non-

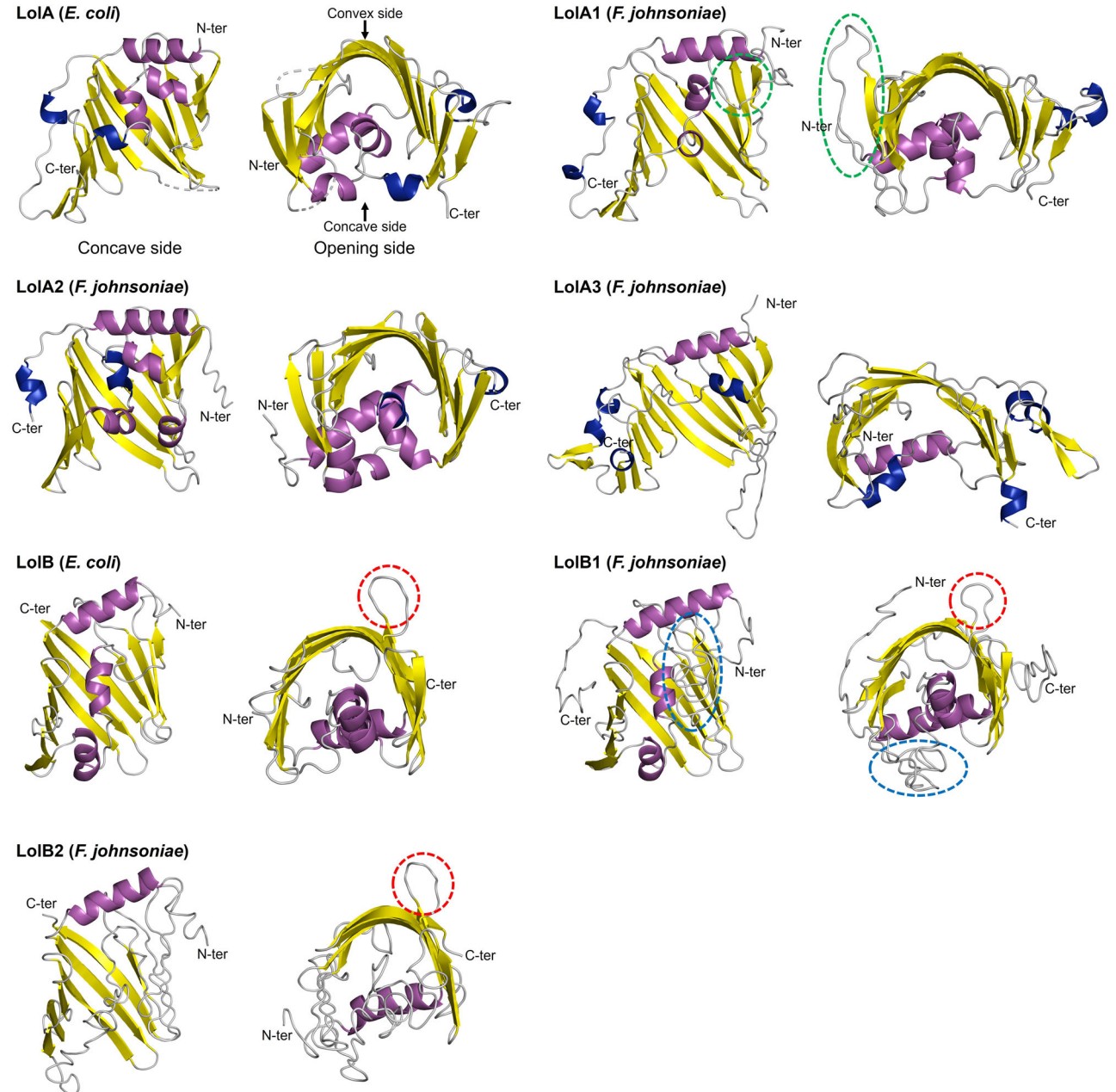

**Fig. 1 | Structural comparison between *E. coli* and *F. johnsoniae* LolA and LolB homologs.** Comparison of crystallized *E. coli* LolA (PDB entry: 1UA8) and LolB (PDB entry: 1IWM) structures with models of LolA1, LolA2, LolA3, LolB1, and LolB2 homologs from *F. johnsoniae*. Each protein is shown as a cartoon representation and secondary structure elements are colored as follows: α-helix (purple), 3₁₀-helix (blue), β-sheet (yellow), and coil (gray). On each structure, the N- and C-terminal positions are indicated. In LolA1, the loop extension between α-helix 3 and β-strand 7 is highlighted with a dashed green ellipse. In LolB, LolB1, and LolB2, the upward conserved loop is highlighted with a dashed red circle. In LolB1, the compacted loop extension between α-helix 3 and β-strand 7 is highlighted with a dashed blue ellipse.

gliding mutants that had transposon insertions in *lolA* homologs[18,23]. The *F. johnsoniae* gene identified was *Fjoh_2111*, which we refer to as *lolA1*. Since LolA allows lipoproteins to cross the periplasm and localize to the OM, the lack of gliding of the Tn mutants could be due to mislocalization and consequent depletion of lipoproteins in the OM. To confirm the absence of motility reported for a *lolA1* mutant and to determine whether deletion of any of the other *F. johnsoniae* LolA and LolB proteins could have a similar effect, we assessed the gliding motility of the *lolA1*, *lolA2*, *lolA3*, *lolB1*, and *lolB2* mutants on plates. As expected, *lolA1* deletion resulted in non-spreading colonies, and, a lack of motility comparable to that of a *gldJ* non-gliding mutant was also observed when *lolB1* was deleted[24] (Fig. 2a and Supplementary Movie 1). GldJ is an OM lipoprotein required for both

gliding and T9 secretion[24,25]. Deletion of *lolA2*, *lolA3*, and *lolB2* did not affect gliding. Co-deletion of both *lolA1* and *lolB1* resulted in the same phenotype as the single *lolA1* mutant. Plasmid-borne complementation of *lolA1* and *lolB1* deletion with *lolA1* and *lolB1* fully restored gliding motility (Fig. 2a).

Since gliding motility and T9 secretion are highly interlinked and both require several OM lipoproteins for their activity[8,26], we next determined the impact of *lolA* and *lolB* homologs deletion on T9SS activity. This secretion system is unique to Bacteroidota and is involved in the secretion of various proteins mainly involved in adhesion and nutrient acquisition, such as α-amylases that allow *F. johnsoniae* to hydrolyze and feed on starch[25]. To monitor Type 9 secretion, we measured the starch degradation activity of

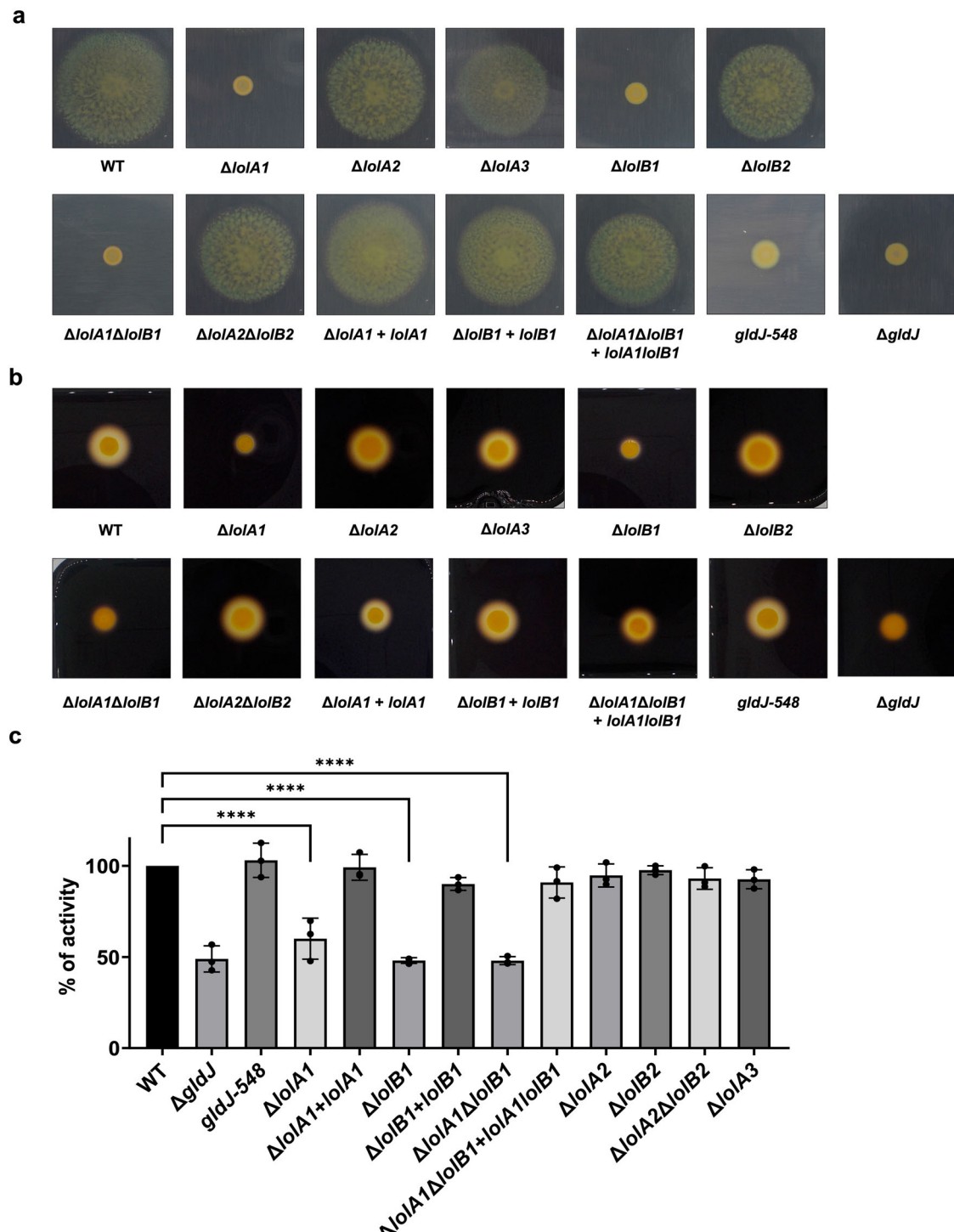

**Fig. 2 | Deletion of *lolA1* and *lolB1* affects gliding motility and Type IX secretion.** Gliding motility on MM agar plates after 48 h (**a**). Starch degradation by T9-secreted amylases on LB starch plates after 24 h (**b**) and amylase activity of cell culture supernatants (**c**). Data in (**c**) is represented as means of *n* = 3 independent biological replicates ± standard deviation, and statistical analysis was a one-way ANOVA followed by Dunnett's multiple comparison test ($F$ (22, 46) = 47.47). Asterisks indicate the following: (****): $p < 0.0001$.

secreted amylases on starch-containing plates and on cell culture supernatants. Starch degradation was significantly reduced in the *lolA1*, *lolB1*, and *lolA1lolB1* mutant strains compared to the WT and like that of the *gldJ* mutant (Fig. 2b, c). Deletion of *lolA2*, *lolA3*, and *lolB2* did not affect amylase secretion (Fig. 2b, c). Plasmid-borne complementation of *lolA1* and *lolB1* strains fully restored amylase secretion (Fig. 2b, c). The reduced amylase activity observed for the *lolA1* and *lolB1* mutants could not be ascribed to the lack of motility of these strains as a *gldJ-548* mutant, which is non-motile but

has an active T9SS[8], degraded starch to the same extent as the WT strain (Fig. 2b, c).

Overall, these data indicate that LolA1 and LolB1 are crucial for both gliding motility and T9 secretion and reinforce our initial hypothesis that these two proteins might belong to the same pathway, namely the transport of a subset of lipoproteins to the OM. Given their pivotal role in *F. john-soniae*, we focused on further characterizing the function of these two proteins.

## a

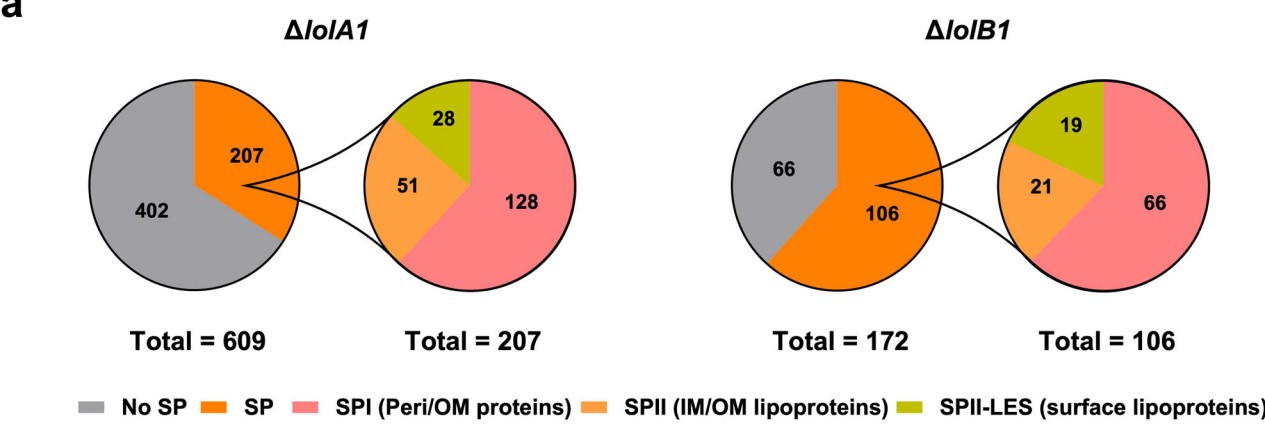

**ΔlolA1** — Total = 609, Total = 207

**ΔlolB1** — Total = 172, Total = 106

Legend: No SP, SP, SPI (Peri/OM proteins), SPII (IM/OM lipoproteins), SPII-LES (surface lipoproteins)

## b

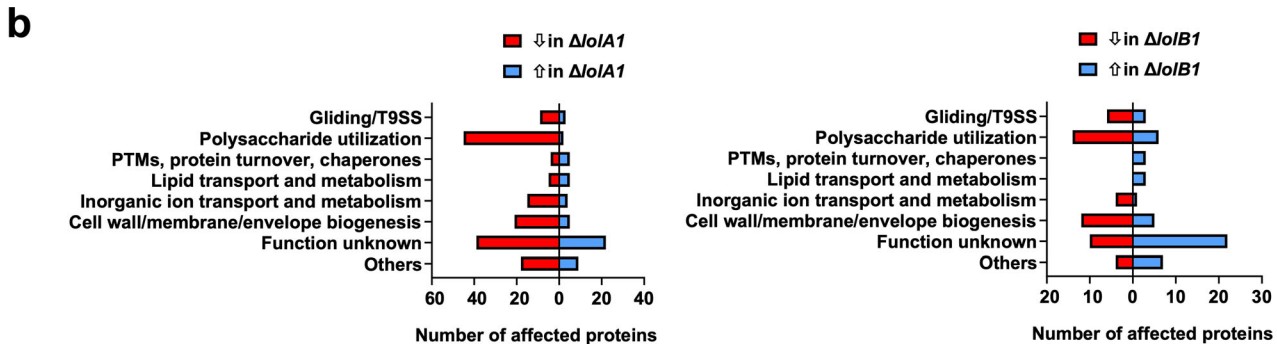

Left chart: ⇓ in ΔlolA1 (red), ⇑ in ΔlolA1 (blue)

Right chart: ⇓ in ΔlolB1 (red), ⇑ in ΔlolB1 (blue)

Categories: Gliding/T9SS, Polysaccharide utilization, PTMs, protein turnover, chaperones, Lipid transport and metabolism, Inorganic ion transport and metabolism, Cell wall/membrane/envelope biogenesis, Function unknown, Others

Number of affected proteins

## c

| Description | Signal peptide | Accession | Gene code | ΔlolA1/WT FC | ΔlolB1/WT FC |
|---|---|---|---|---|---|
| SprD | SPI | A1E5U4 | *Fjoh_0980* | 0.05 | - |
| GldJ | SPII | A5FJM9 | *Fjoh_1557* | 0.05 | 0.22 |
| SprF-like protein | SPI | A5FE72 | *Fjoh_3477* | 0.15 | 0.48 |
| SprF-like protein | SPI | A5FCV6 | *Fjoh_3951* | 0.16 | 0.16 |
| SprF | SPI | A5FLA7 | *Fjoh_0978* | 0.37 | - |
| SprT | SPI | A5FJX0 | *Fjoh_1466* | 0.41 | 1.67 |
| GldK | SPII | A5FIS8 | *Fjoh_1853* | 0.48 | 0.08 |
| GldN | SPI | A5FIT1 | *Fjoh_1856* | 0.53 | 0.33 |
| SprB | SPI | A1E5U5 | *Fjoh_0979* | - | 0.32 |
| SprF-like protein | SPI | A5FJB8 | *Fjoh_1677* | - | 1.85 |
| RemH | SPI | A5FL98 | *Fjoh_0984* | - | 1.78 |
| RemF | SPI | A5FEZ9 | *Fjoh_3206* | - | 1.93 |
| GldH | SPII | Q8KRP0 | *Fjoh_0890* | 2.31 | - |
| PorU | SPI | A5FJM8 | *Fjoh_1556* | - | 1.51 |
| RemI | SPI | A5FF07 | *Fjoh_3194* | 2.72 | 1.66 |

**Fig. 3 | The OM protein composition of *lolA1* and *lolB1* mutants is altered compared to the WT.** Proteins identified by mass spectrometry whose abundance significantly differs (FC ≥ 1.5, significance ≥ 20) in the OM fraction of the *lolA1* and *lolB1* deleted strains compared to the WT clustered by predicted localization (SignalP server). SPI signal peptide I (OMP or periplasmic), SPII periplasmic-facing lipoproteins, SPII-LES surface-exposed lipoproteins (**a**). Proteins with an SP whose abundance is affected in the OM fraction of the *lolA1* and *lolB1* mutants compared to the WT clustered by functional class (EggNOG) (**b**). Gliding and T9SS-specific proteins whose abundance is altered in *lolA1* and/or *lolB1* OM compared to the WT (**c**).

## Deletion of *lolA1* and *lolB1* affects outer membrane proteome composition

The absence of gliding and T9 secretion could be due to mislocalization of OM lipoproteins crucial for these pathways in the absence of LolA1 and LolB1. To test this hypothesis, we determined the impact of *lolA1* and *lolB1* deletion on the OM protein composition of bacteria grown in CYE-rich medium. We purified the OM of WT, *lolA1*, and *lolB1* strains and identified proteins by mass spectrometry. Overall, the absence of *lolA1* or *lolB1* respectively affected 609 and 171 proteins in a significant manner (fold change ≥ 1.5; significance ≥ 20) (Fig. 3a, Supplementary Data 1 and 2). We sorted the proteins according to their localization: cytoplasmic (no SP), integral OM and soluble periplasmic proteins (SPI), periplasmic-facing lipoproteins (SPII), and surface-exposed lipoproteins (SPII-LES). The OM proteome of the WT and *lolB1* mutant was enriched in membrane proteins

and lipoproteins while, that of the *lolA1* mutant contained many cytoplasmic proteins, accounting for 66% of significant entries (402 out of 609) hinting at possible intracellular protein leakage as also suggested by the extracellular debris and cell material observed by electron microscopy for this mutant (Fig. 3a, Supplementary Data 2 and Supplementary Fig. 3). We clustered proteins harboring an SP by predicted function and mapped those involved in polysaccharide utilization to their respective(s) polysaccharide utilization loci (PULs)[27].

In both mutants, the most affected known cellular functions were polysaccharide utilization (2 proteins increased (+) and 45 decreased (−) in *lolA1*, +6/−14 in *lolB1*) and cell wall/membrane/envelope biogenesis (+5/−21 in *lolA1*, +5/−12 in *lolB1*). In addition, several gliding/T9 secretion-specific proteins were affected (+3/−9 in *lolA1*, +3/−6 in *lolB1*). Overall, 207 SPI/SPII proteins were impacted by the deletion of *lolA1* and 106 by the deletion of *lolB1* (Fig. 3b). Among these, the abundance of 59 proteins was decreased in both mutants and among them several gliding/T9 secretion-related proteins (Fig. 3c). GldK and GldJ, two lipoproteins essential for gliding motility and T9 secretion[28] were much less abundant in the OM of the *lolA1* and *lolB1* mutants than in the WT strain (GldK FC in *lolA1* = 0.48 and 0.08 in *lolB1*; GldJ FC in *lolA1* = 0.05 and 0.22 in *lolB1*). GldN, another critical component of the gliding and T9 types of machinery and the physical partner of GldK[29,30] was also less abundant (FC = 0.53 in *lolA1* and 0.33 in *lolB1*). One SprF-like protein (Fjoh_3951), implicated in T9 secretion[28], was also less abundant in the OM of both mutants (FC = 0.16 in *lolA1* and *lolB1*). We also noticed a few proteins potentially involved in envelope biogenesis whose amount in the OM was altered in the two mutants, namely a peptidoglycan-binding LysM protein (FC = 3.12 in *lolA1* and 1.86 in *lolB1*), a BamB-like protein (FC = 0.44 in *lolA1* and 0.57 in *lolB1*), a polysaccharide export protein Wza (FC = 0.27 in *lolA1* and 0.23 in *lolB1*), and a peptidoglycan hydrolase (FC = 0.18 in *lolA1* and 0.28 in *lolB1*).

Among the 148 proteins specifically affected by the deletion of *lolA1*, we found other components of the gliding/T9 secretion types of machinery (Fig. 3c). Among them, SprT (FC = 0.41), SprF (FC = 0.37), and SprD (FC = 0.05) are all required for gliding motility[31,32]. The absence of *lolA1* resulted in the decrease of numerous OM proteins dedicated to polysaccharide utilization and inorganic ion transport (Fig. 3b) of which most were SusC-like proteins or TonB-dependent receptors, i.e., proteins with a β-barrel domain.

Within the 47 proteins specifically affected by the deletion of *lolB1*, we also found gliding/T9 secretion-related proteins (Fig. 3c). LolA1 amount was also increased in the *lolB1* mutant (FC = 2.04). In contrast with the data specific to *lolA1*, we did not observe an obvious pattern in the data specific to *lolB1*. The loss of *lolB1*, as for *lolA1*, decreased the number of PUL-encoded proteins in the OM but to a much lower extent (Fig. 3b).

We did not observe a general mislocalization of OM lipoproteins upon deletion of *lolA1* or *lolB1*, as one could have expected. Nevertheless, the proteomics data confirm and provide an explanation for the loss of gliding motility and T9 secretion that we observed for both mutants. Although the deletion of *lolA1* or *lolB1* alters the OM protein composition in different ways, bacteria lacking either of these two proteins fail to properly localize a subset of proteins and lipoproteins, including core components of gliding motility/T9 secretion such as GldN, GldK, and GldJ. While the deletion of *lolA1* was more consequential for the OM of *F. johnsoniae* than the deletion of *lolB1*, the most impacted cellular functions (polysaccharide utilization and cell wall/membrane/envelope biogenesis) were in common. Considering that the abundance of several OM lipoproteins was altered in both mutants, our data suggest that *lolA1* and *lolB1* participate in the same pathway, i.e., the transport of lipoproteins to the OM and, in particular, lipoproteins involved in gliding motility and T9 secretion. Yet, *lolA1* deletion clearly determined a bigger perturbation of the OM proteome. Among the IM and cytoplasmic proteins identified in the proteomic analysis, the majority were more abundant in the *lolA1* (323 out of 402) and *lolB1* (38 out of 66) mutants OM compared to the WT, thus pointing to some envelope defects of these mutants, probably more pronounced in the absence of LolA1 (Supplementary Data 2).

## Deletion of *lolA1* and *lolB1* affects growth and cell morphology

The OM proteome composition alterations observed prompted us to test whether the lack of LolA1 and LolB1 would affect bacterial fitness. To this aim we monitored the growth of these mutants in two media, casitone yeast extract (CYE), the rich medium normally used to grow *F. johnsoniae* and motility medium (MM) a poor medium that stimulates gliding motility[33].

As shown in Fig. 4a, all mutants grew like the WT in CYE. In contrast, in MM, the deletion of *lolA1* severely impacted growth while the deletion of *lolB1* did not (Fig. 4a). In the absence of both LolA1 and LolB1 growth was comparable to that of the single *lolA1* mutant. Plasmid-borne complementation of *lolA1* and *lolA1lolB1* mutants with *lolA1* and *lolA1* and *lolB1* fully restored growth to the WT levels (Fig. 4a). Deletion of any of the other LolA and LolB proteins did not affect growth in any condition tested (Fig. 4a).

Next, we assessed whether the mutations had any morphological effect on bacteria. We observed by optic and transmission electron microscopy cells grown for 16 h in CYE medium and found that *lolA1* bacteria presented some aberrant shapes, some resembling the shape of a lollipop, while others being completely round (Fig. 4b and Supplementary Fig. 3). The same abnormal morphologies, along with more cell aggregation, were also observed in the *lolA1lolB1* double mutant, while *lolB1* mutants resembled the WT strain. Both *lolA1* and *lolA1lolB1* mutants displayed detached OM and release of intracellular material as a result of severe envelope defects (Supplementary Fig. 3). The aberrant cell morphology phenotype of the *lolA1* and *lolA1lolB1* strains was enhanced when bacteria were grown in MM and the formation of abnormal cells was also observed for the *lolB1* mutant in this medium (Fig. 4b). Lack of any of the other LolA and LolB proteins did not affect cell morphology in either CYE or MM (Supplementary Fig. 4). Complementation of *lolA1, lolB1,* and *lolA1lolB1* with plasmid-borne *lolA1, lolB1,* and *lolA1* and *lolB1* fully restored the cell morphology to WT levels (Fig. 4b).

Overall, while growth in the MM medium was affected only by the deletion of *lolA1*, the deletion of *lolA1* and *lolB1* affected cell morphology, in particular in MM for this latter mutant. Co-deletion of *lolA1* and *lolB1* determined an enhancement in abnormal cell morphology in both CYE and MM (Fig. 4b and Supplementary Fig. 3).

## Deletion of *lolA1* and to a minor extent of *lolB1* affects envelope integrity

The observed growth and morphological phenotypes of the *lolA1* mutant when grown in MM hinted to a possible membrane instability. A main difference in composition between these two media is the absence of magnesium in the form of $MgSO_4$ in MM. Divalent cations and, in particular, magnesium, are known to stabilize LPS–LPS interactions[34]. In bacteria with OM alterations due to protein and/or lipid shifts, magnesium has been shown to compensate for these phenotypes. We thus tested whether the growth defect and morphology changes observed for the *lolA1* and *lolB1* mutants could be rescued by magnesium supplementation in MM. As shown in Fig. 5, the addition of 8 mM $MgSO_4$ (as in CYE) to MM restored growth (Fig. 5a) of the *lolA1* and *lolA1lolB1* mutants, thus suggesting an alteration of the OM of these mutants. Concerning cell morphology, the addition of magnesium alleviated abnormal cell morphologies of the mutants but did not totally restore them to WT levels (Fig. 5b).

We also tested whether *lolA1* and/or *lolB1* deletion might affect the sensitivity of the mutants to several OM perturbing agents, namely SDS, Polymyxin B, and EDTA. We observed that while deletion of *lolA1* and/or *lolB1* did not affect growth at low SDS concentrations, the *lolA1* mutant showed significant growth impairment at a higher concentration (Fig. 5c). The *lolB1* strain also showed reduced growth although to a lesser extent than the *lolA1* mutant (Fig. 5c). The *lolA1lolB1* double mutant showed a phenotype like the *lolA1* mutant. Polymyxin B affected the growth of the three mutants in a similar concentration-dependent fashion (Fig. 5c). In contrast, EDTA affected the growth of the *lolA1* and *lolA1lolB1* mutants, but not of the *lolB1* mutant at the tested concentrations (Fig. 5c).

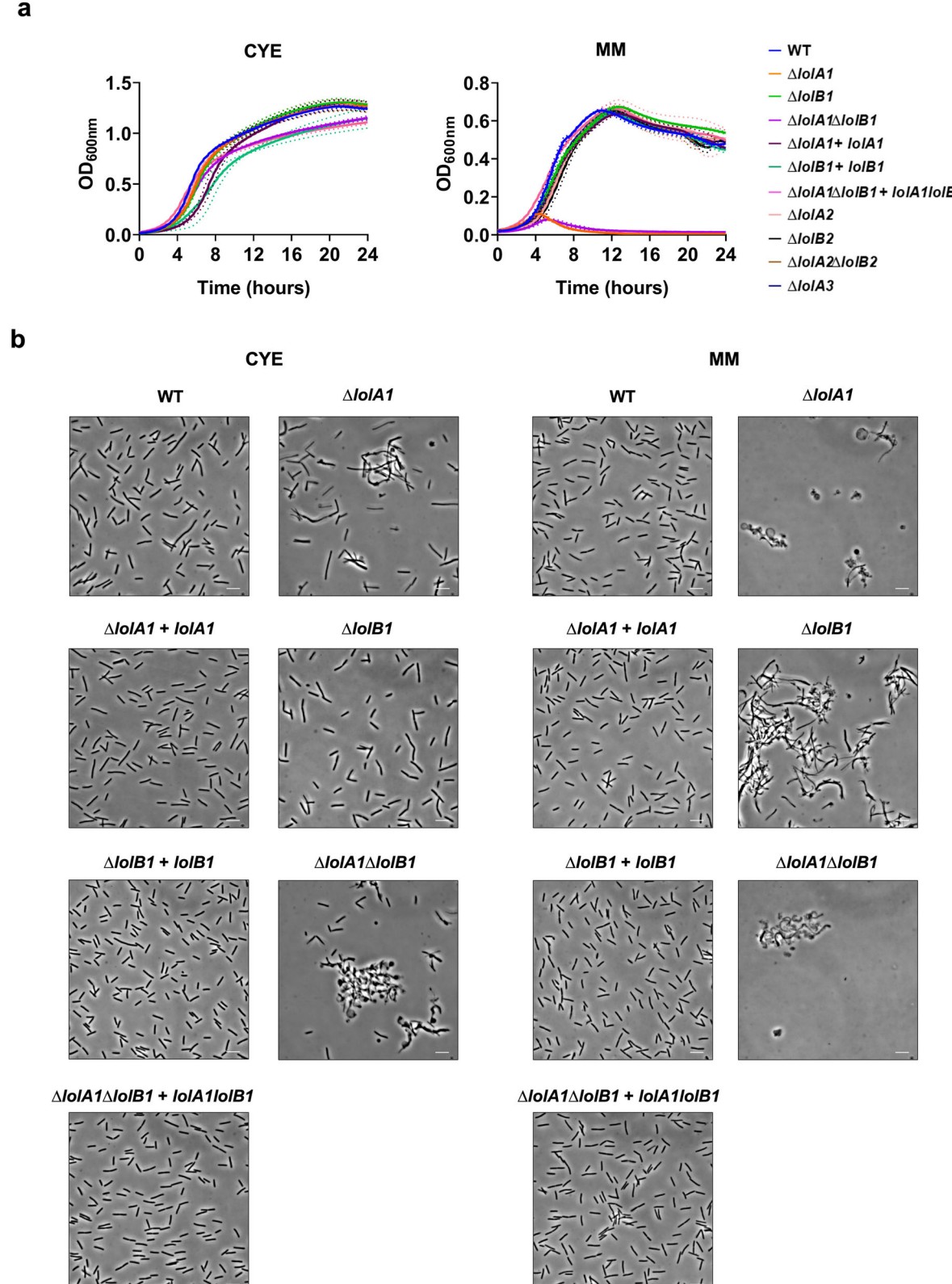

**Fig. 4 | Deletion of *lolA1* and *lolB1* affects growth and cell morphology.** Growth curves of WT and mutant strains in CYE and MM liquid media. Data are represented as means of *n* = 3 independent biological replicates ± standard deviation (**a**). Phase-contrast microscopy images of bacteria grown for 16 h in CYE and MM media (bar = 5 μm) (**b**).

At an EDTA concentration of 1 mM, the *lolA1lolB1* double mutant showed an increased sensitivity compared to that of the *lolA1* mutant, meaning that the deletion of *lolB1* in a *lolA1* mutant background increases the sensitivity of bacteria to this compound.

Taken together, these results suggest that the deletion of *lolA1* affects OM stability, as the *lolA1* and *lolA1lolB1* mutants showed acute sensitivity to all tested stresses. On the other hand, *lolB1* single deletion seems to have a lower impact on OM stability, as this mutant was less sensitive than the

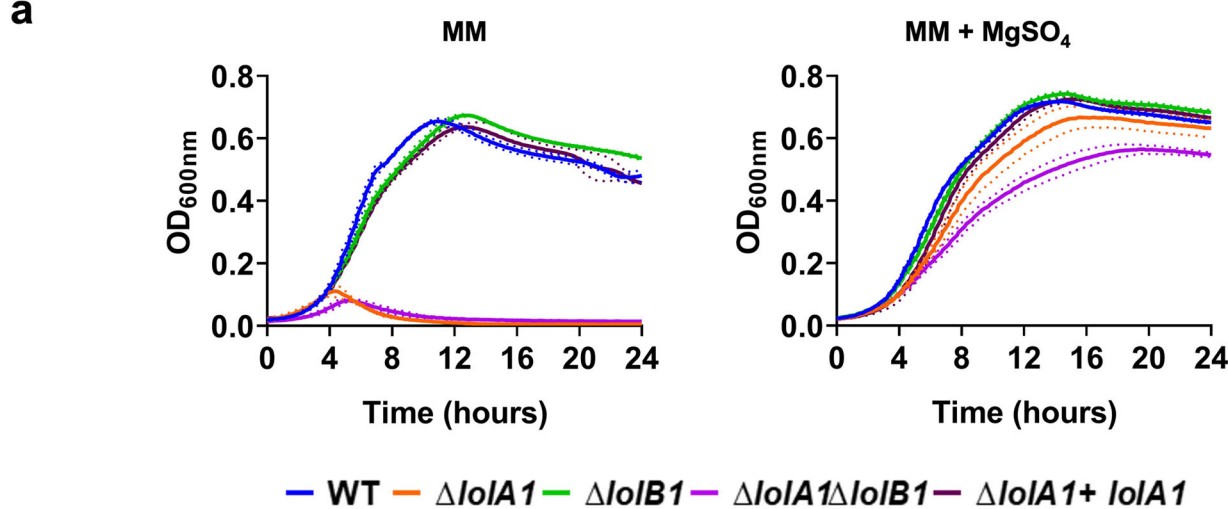

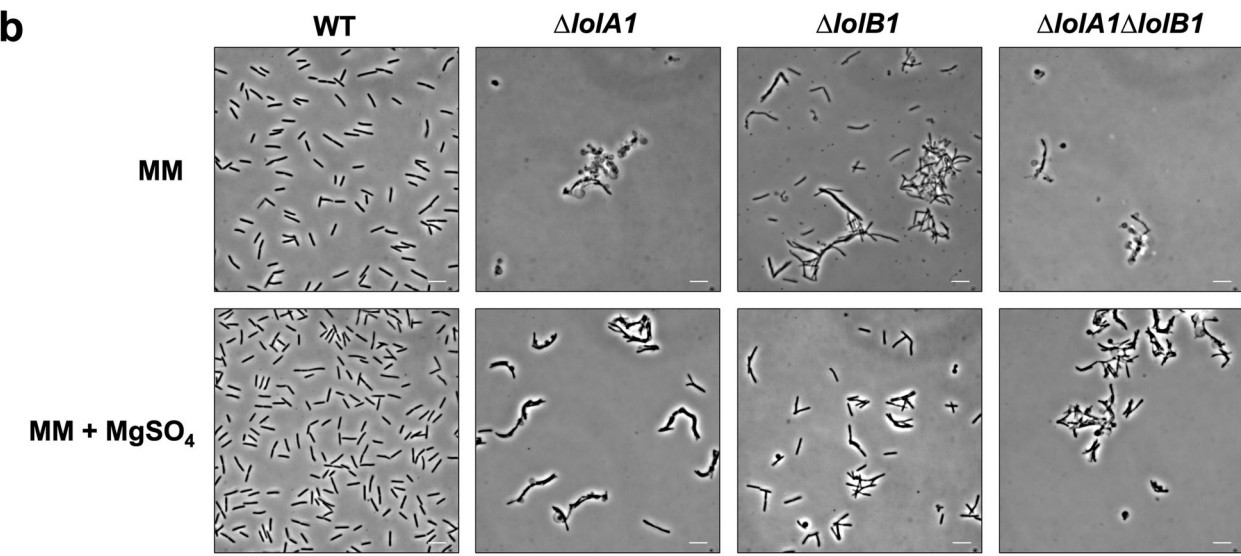

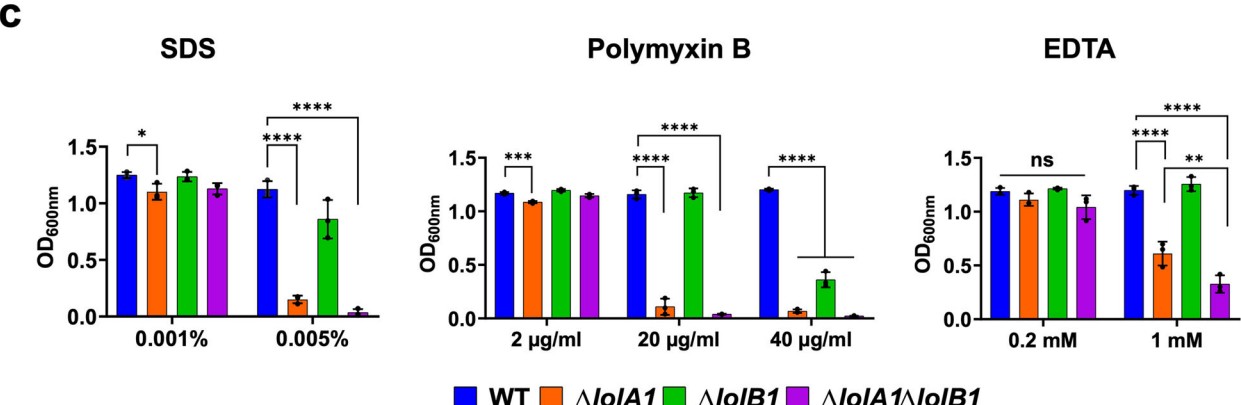

**Fig. 5 | Impact of deletion of *lolA1* and of *lolB1* on OM integrity and stress sensitivity.** Growth curves of WT and mutant strains in MM liquid medium without and with 8 mM $MgSO_4$. Data are represented as means of $n = 3$ independent biological replicates ± standard deviation (**a**). Phase-contrast microscopy images of bacteria grown in liquid MM medium without and with 8 mM $MgSO_4$ (bar = 5 μm) (**b**). Growth after 24 h in CYE liquid medium in the presence of different OM perturbing agents (SDS, Polymyxin B, and EDTA) (**c**). Data in **c** are represented as means of $n = 3$ biological independent replicates ± standard deviation, and statistical analysis was a one-way ANOVA followed by Tukey's multiple comparison test (SDS 0.001% $F_{(3, 8)} = 6.602$; SDS 0.005% $F_{(3, 8)} = 440.9$; Polymyxin B 2 μg/ml $F_{(3, 8)} = 43.21$; Polymyxin B 20 μg/ml $F_{(3, 8)} = 540.5$; Polymyxin B 40 μg/ml $F_{(3, 8)} = 669.9$; EDTA 0.2 mM $F_{(3. 8)} = 4.37$; EDTA 1 mM $F_{(3, 8)} = 99.03$). Asterisks indicate the following: (*): $p < 0.0332$, (**): $p < 0.0021$, (***): $p < 0.0002$, (****): $p < 0.0001$, (ns): $p > 0.05$.

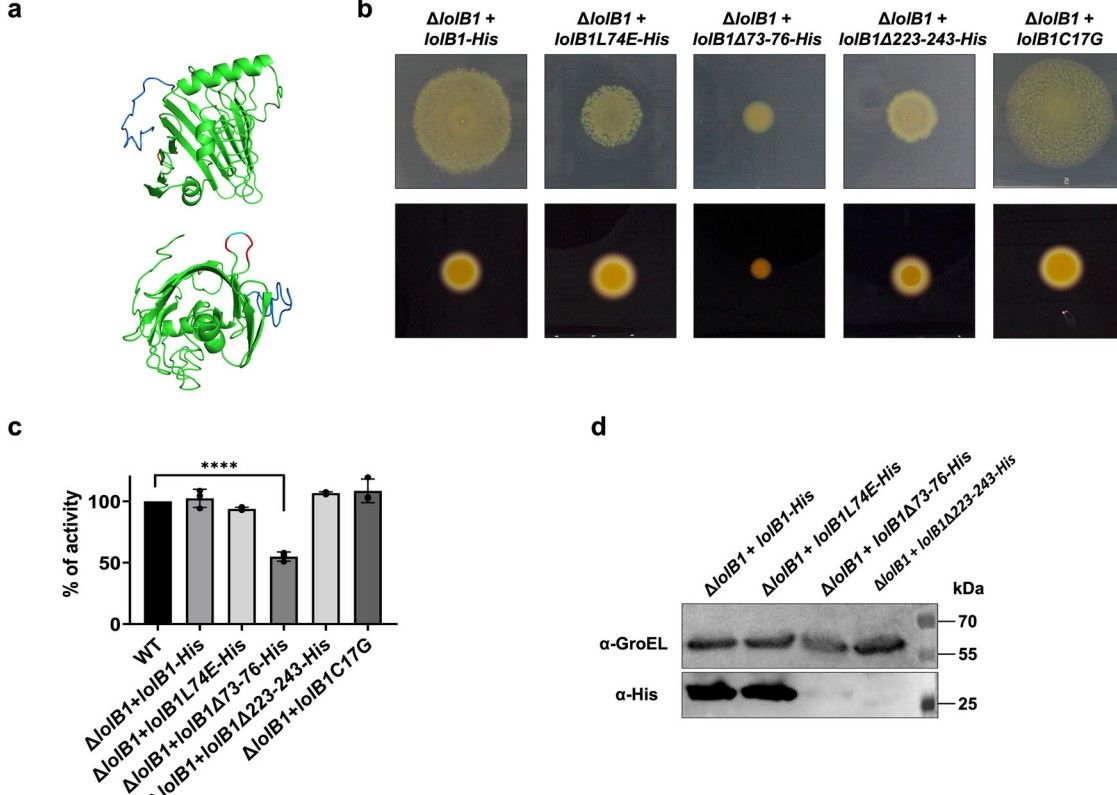

**Fig. 6 | The protruding loop and the C-terminal domain of LolB1 are crucial for its function.** I-TASSER model of LolB1 with the protruding loop (in red) with the conserved L74 (in cyan) and the C-terminal domain (in blue) (**a**). Gliding on MM agar plates after 48 h and starch degradation on LB starch plates after 24 h of *lolB1* mutants expressing: WT LolB1, LolB1$_{L74E}$, LolB1$_{\Delta73-76}$, LolB1$_{\Delta223-243}$ and LolB1$_{C17G}$ (**b**). Amylase activity of the supernatant of WT and *lolB1* mutants expressing: WT LolB1, LolB1$_{L74E}$, LolB1$_{\Delta73-76}$, LolB1$_{\Delta223-243}$ and LolB1$_{C17G}$ (**c**). Data in **c** are represented as means of $n = 3$ independent biological replicates ± standard deviation, and statistical analysis was a one-way ANOVA followed by Dunnett's multiple comparison test ($F$ (22, 46) = 47.47). Asterisks indicate the following: (****): $p < 0.0001$. Detection by Western blot of C-terminally His-tagged: WT LolB1, LolB1$_{L74E}$, LolB1$_{\Delta73-76}$ and LolB1$_{\Delta223-243}$. GroEL protein levels are shown and serve as a loading control (**d**).

*lolA1* mutant to SDS and Polymyxin B and was resistant to EDTA. The increased sensitivity of the *lolA1lolB1* double mutant to EDTA compared to the *lolA1* single mutant supports our observations of cell morphology (Fig. 4b) as well as the idea that the deletion of both *lolA1* and *lolB1* is more detrimental to *F. johnsoniae* than their single deletion.

### The protruding loop and the C-terminal region of LolB1 are crucial for its function

In *E. coli* LolB, the hydrophobic protruding loop between β-strands 3 and 4 is critical for lipoprotein insertion[35]. A highly conserved residue, Leu68, found at the tip of the loop has been shown to be crucial for LolB activity. When this leucine is replaced with a polar residue or deleted, LolB can no longer efficiently insert lipoproteins into the OM[13,35]. A loop is also present in LolB1 with a leucine (L74) at its tip (Fig. 6a). To see whether the loop and this residue are crucial for *F. johnsoniae* LolB1 function, we tested whether the expression of LolB1 where Leu74 is replaced by the polar glutamate (L74E) or where the loop is deleted (residues 73–76) could complement the gliding and T9SS phenotypes of the *lolB1* mutant. As shown in Fig. 6b, c, the LolB1$_{L74E}$ variant only partially restored gliding motility whereas amylase secretion was restored to WT levels. In contrast, deletion of the loop completely abolished gliding and secretion.

These data suggest that the loop and the conserved Leu are important for LolB1 function, as in *E. coli* LolB. Partial complementation of the LolB1$_{L74E}$ protein could not be ascribed to a different protein expression level of this mutation since the protein amount of WT LolB1 and LolB1$_{L74E}$ variants was comparable (Fig. 6d). In contrast, deletion of residues 73–76 of LolB1 (LolB1$_{\Delta73-76}$) resulted in a significant reduction of protein amount,

as shown by the weak signal in the western-blot and by MS detection (Fig. 6d and Supplementary Table 2), thus suggesting that this mutation affects protein stability or induces degradation of this non-functional LolB variant.

A main difference between *E. coli* LolB and LolB1 is the presence of a C-terminal domain that could not be folded by modeling software (Fig. 6a). We wondered whether this domain could be important for LolB1 function and thus deleted it (*lolB1Δ223–243*). This mutant LolB1 could only partially complement gliding while amylase secretion was like WT levels (Fig. 6b, c). As for LolB1$_{\Delta73-76}$, deletion of the C-terminal domain determined a significant reduction in the protein levels, as shown by the weak signal in the western blot and low spectral count detected by MS (Fig. 6d and Supplementary Table 2). However, the evidence that deletion of the C-terminal tail of LolB1 still allows protein secretion despite determining the strongest reduction in protein levels (Supplementary Table 2), suggests that this deletion affects protein stability but not completely protein function as observed for the LolB1 73–76 residues deletion which abolished both gliding and secretion.

In *E. coli*, LolB does not require its lipid anchor to correctly insert lipoproteins in the OM[36]. To see if this was also the case for LolB1, we generated a periplasmic soluble form of LolB1, by introducing a C17G mutation in the signal peptide of LolB1 (LolB1$_{C17G}$), thus replacing the lipidated cysteine and generating an SPI. In trans expression of LolB1$_{C17G}$ fully complemented *lolB1* deletion in *F. johnsoniae* as shown by the recovered gliding motility and amylase secretion (Fig. 6b, c). As for *E. coli* LolB, lipidation does not seem to be crucial for *F. johnsoniae* LolB1 function.

**Fig. 7 | *E. coli* LolA and LolB do not complement *lolA1* and *lolB1* deletion in *F. johnsoniae*.** Gliding motility on MM agar plates after 48 h (**a**) and starch degradation on LB starch plates after 24 h (**b**). Amylase activity of cell culture supernatants (**c**). Data in **c** are represented as means of $n = 3$ independent biological replicates ± standard deviation, and statistical analysis was a one-way ANOVA followed by Dunnett's multiple comparison test ($F$ (22, 46) = 47.47). Asterisks indicate the following: (****): $p < 0.0001$.

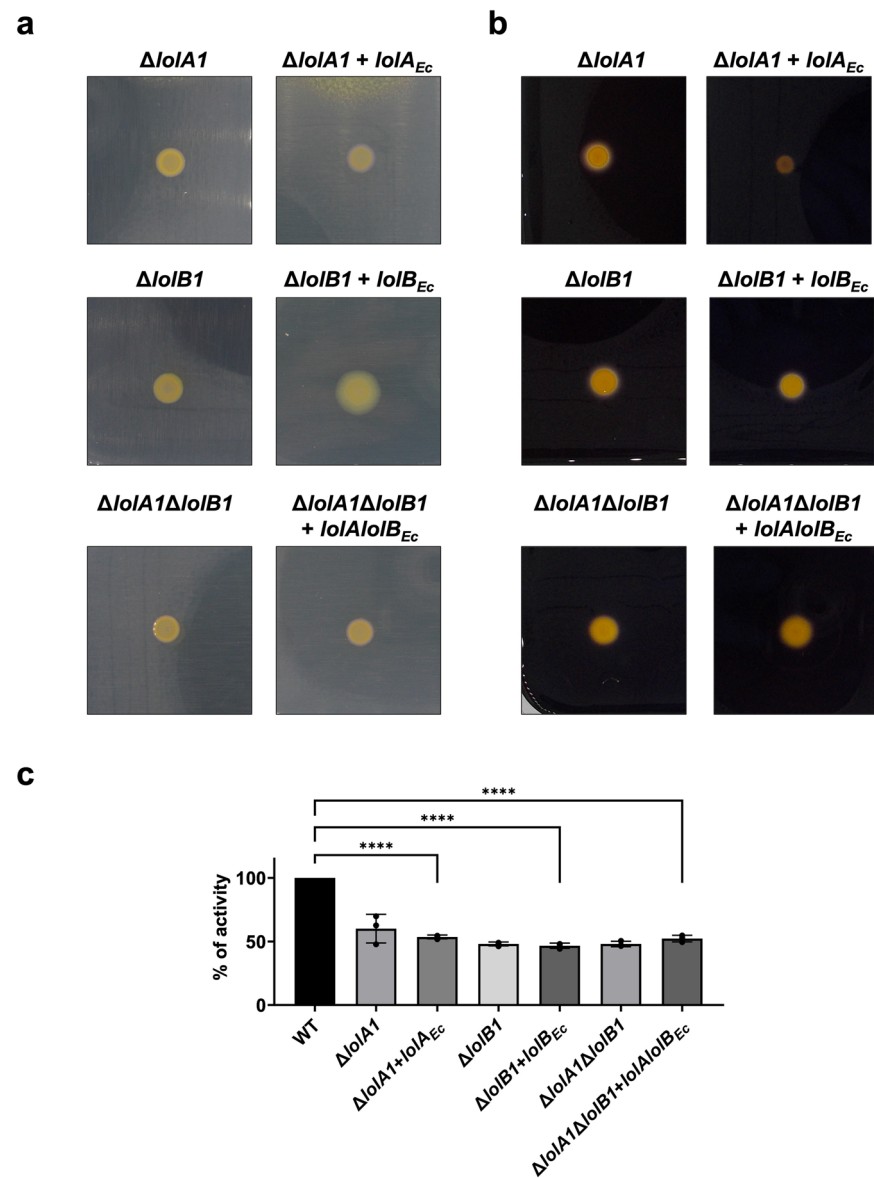

## *E. coli* and *F. johnsoniae* LolA and LolB are not interchangeable

In silico structure comparison and protein characterization show high structural similarity between *F. johnsoniae* LolA1 and LolB1 and *E. coli* LolA and LolB respectively (Fig. 1) while highlighting some differences regarding hydrophobicity and charge distribution (Supplementary Figs. 1 and 2 and Supplementary Table 1). We thus wondered whether LolA and LolB from *E. coli* might complement the deletion of *lolA1* and *lolB1* in *F. johnsoniae*. In trans expression of LolA$_{Ec}$ and LolB$_{Ec}$ did not restore gliding nor T9 secretion of the *lolA1* and *lolB1* mutants (Fig. 7a–c). Similarly, co-expression of LolA$_{Ec}$ and LolB$_{Ec}$ did not restore gliding nor secretion of the *lolA1lolB1* double mutant (Fig. 7a–c).

Next, we tested whether *F. johnsoniae* LolA1 and LolB1 could complement the deletion of *lolA* and *lolB* in *E. coli*. Since *lolA* and *lolB* are essential genes in *E. coli*, we first expressed the *F. johnsoniae* proteins LolA1 and LolB1 individually or together (LolA1–LolB1) in the *E. coli* MG1655 WT strain and then attempted to delete *lolA* or *lolB*. While deletion of *lolA* and *lolB* could be achieved when LolA$_{Ec}$ and LolB$_{Ec}$ were expressed in trans

(Supplementary Fig. 5), we could not obtain any mutant when the *F. johnsoniae* proteins were expressed. Similar results were obtained when LolA2 and/or LolB2 proteins were expressed in *E. coli*. Lack of complementation by *E. coli* LolA and LolB of *F. johnsoniae* mutants and by *F. johnsoniae* LolA1 and LolB1 of *E. coli* mutants could not be ascribed to lack of expression of these proteins since all of them could be detected by mass spectrometry analysis (Supplementary Table 3). In conclusion, despite structural and physico-chemical similarities, *E. coli* and *F. johnsoniae* LolA and LolB proteins are not interchangeable.

## LolA and LolB are conserved among Bacteroidota

We next explored if the *F. johnsoniae* LolA and LolB proteins were conserved in bacteria of the phylum Bacteroidota. To this aim, we performed an in silico sequence similarity search for homologs of the five *F. johnsoniae* LolA and LolB proteins in several Bacteroidota species. This analysis identified LolA and LolB homologs in all the analyzed species (Fig. 8 and Supplementary Table 4). In addition, while we found LolA1 and LolB1 homologs in all species, LolA2, LolB2, and LolA3 homologs seem to be more restricted. Among the species carrying only LolA1 and LolB1 homologs, there is *C. canimorsus*, which belongs to the family Flavobacteriaceae and is a normal oral commensal of dogs and cats which can cause severe infections

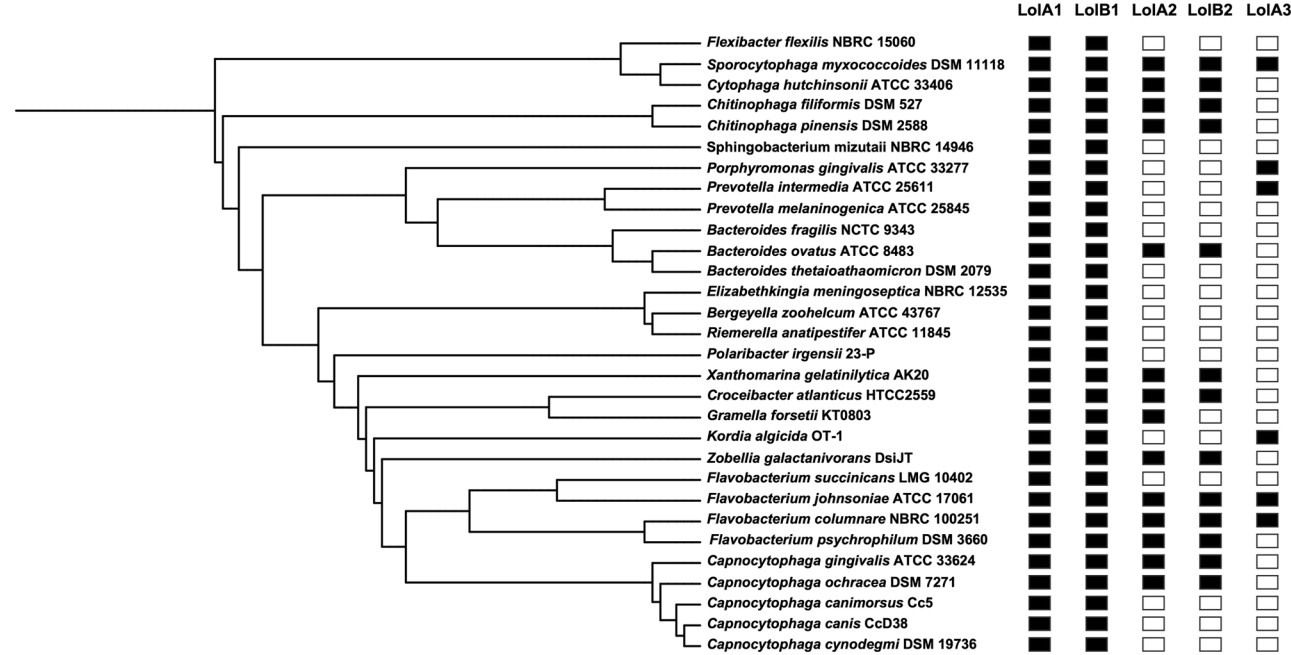

**Fig. 8 | Conservation of LolA and LolB in Bacteroidota.** Homologs of *F. johnsoniae* LolA and LolB proteins in several Bacteroidota species identified by DELTA Blast[62]. Phylogenetic tree based on NCBI taxonomy generated via phyloT[67].

in humans upon contact with these animals[37]. In *C. canimorsus*, LolA and LolB homologs, Ccan_16490 (LolA) and Ccan_17050 (LolB), are essential since attempts to delete their encoding genes failed. Regarding *lolA2* and *lolB2*, both genes were always found to be encoded in loci dedicated to the synthesis and transport of a flexirubin-like pigments[19], suggesting that LolA2 and LolB2 are specific to these processes.

To see whether the *F. johnsoniae* LolA1 and LolB1 and *C. canimorsus* LolA and LolB homologs share the same function in these bacteria, we first tested whether expression of *C. canimorsus* LolA (Ccan_16490) and LolB (Ccan_17050) could complement the deletion of *lolA1* and *lolB1* in *F. johnsoniae* respectively. Complementation with *C. canimorsus* LolA or LolB fully restored the ability to glide and secrete via the T9SS of the *F. johnsoniae lolA1* and *lolB1* mutants (Fig. 9a, b) as well as the growth and cell morphology phenotypes in MM (Fig. 9c, d).

We next tested whether the expression of *F. johnsoniae* LolA1 and LolB1 could bypass the lethality of the deletion of *lolA* and *lolB* of *C. canimorsus*. To test complementation with LolA1, we performed a plasmid exchange in the *C. canimorsus lolA (Ccan_16490)* mutant strain expressing plasmid-borne *C. canimorsus lolA* with the vector encoding *lolA1* (see "Methods" for additional details). To test LolB1 complementation, we first expressed LolB1 in trans in the *C. canimorsus* WT strain and then deleted *lolB (Ccan_17050)*. Both *C. canimorsus lolA* and *lolB* deletion mutants were viable when LolA1 or LolB1 were expressed, thus confirming that the *F. johnsoniae* and *C. canimorsus* proteins share similar functions (Fig. 9e). However, while complementation of *lolB* deletion in *C. canimorsus* with *lolB1* did not result in any growth defect on rich medium, expression of *lolA1* only partially complemented *lolA* deletion (Fig. 9e). Overall, these results suggest that homologs of *F. johnsoniae* LolA1 and LolB1 share the same function in *C. canimorsus* and that the same could be true in Bacteroidota in general.

### Simultaneous deletion of all *F. johnsoniae* LolA and LolB homologs is not lethal and does not affect surface lipoprotein localization

Next, to see whether *F. johnsoniae* could tolerate the lack of all its LolA and LolB homologs, we generated a *lolA1 lolA2 lolA3 lolB1 lolB2* quintuple mutant by sequentially deleting all genes. This indicates that the bacterium can perform its vital functions even in the absence of any LolA and LolB.

Finally, we tested whether surface-exposed lipoproteins are correctly localized in the absence of all LolA and LolB homologs in *F. johnsoniae*. To this end, we monitored surface lipoprotein exposure using a reporter lipoprotein: the sialidase (SiaC) from *C. canimorsus*. SiaC is an OM lipoprotein facing the periplasm where it is responsible for the cleavage of sialic acid from eukaryotic glycoproteins[38]. The addition of the lipoprotein export signal (LES) to SiaC (LES-SiaC) was shown by our group to be sufficient to localize it at the surface of *C. canimorsus*[16,17].

We thus expressed SiaC and LES-SiaC in the *F. johnsoniae* WT and in the *lolA1 lolA2 lolA3 lolB1 lolB2* quintuple mutant strain and verified their localization by immunofluorescence microscopy using anti-SiaC antibodies and secondary fluorescently labeled antibodies (Fig. 10). As expected, SiaC WT was not exposed at the bacterial surface of both strains while we found LES-SiaC surface exposed in both WT and LolAB lacking strains (Fig. 10), thus suggesting that surface-exposed lipoproteins correctly localize at the cell surface in the absence of any LolA and LolB proteins and thus that the pathway responsible for surface-lipoproteins localization is independent of LolA and LolB homologs of *F. johnsoniae*.

## Discussion

Here, we disclose the presence of LolB homologs in Bacteroidota, indicating that, in contrast to what was thought, the Lol pathway is conserved. In addition, we found that several LolA and LolB can co-exist in some Bacteroidota species such as *F. johnsoniae*, that encodes three LolA and two LolB homologs. We show that deletion of *lolA1* and/or *lolB1* results in the loss of gliding motility and T9 secretion. Proteomic data show that these two phenotypes can be explained by the significant reduction of key proteins and lipoproteins involved in gliding/T9 secretion in the OM of both *lolA1* and *lolB1* mutants. Indeed, the absence of OM lipoproteins GldJ and/or GldK is sufficient to completely abolish gliding and T9 secretion[8,29]. We believe that these lipoproteins might be mislocalized because they require LolA1 and then LolB1 to reach the OM. Two other proteins essential for gliding motility, SprB and SprD, were decreased in *lolA1* and *lolB1*, though the significance did not meet our statistical parameters for both mutants (Fig. 3c and Supplementary Table 5). The OM of the *lolA1* mutant was enriched in GldH (FC = 2.31), a lipoprotein necessary for the stability of GldJ and/or GldK[8,24]. This might be an attempt by *F. johnsoniae* to restore gliding motility and T9 secretion by stabilizing GldJ and GldK within the machinery.

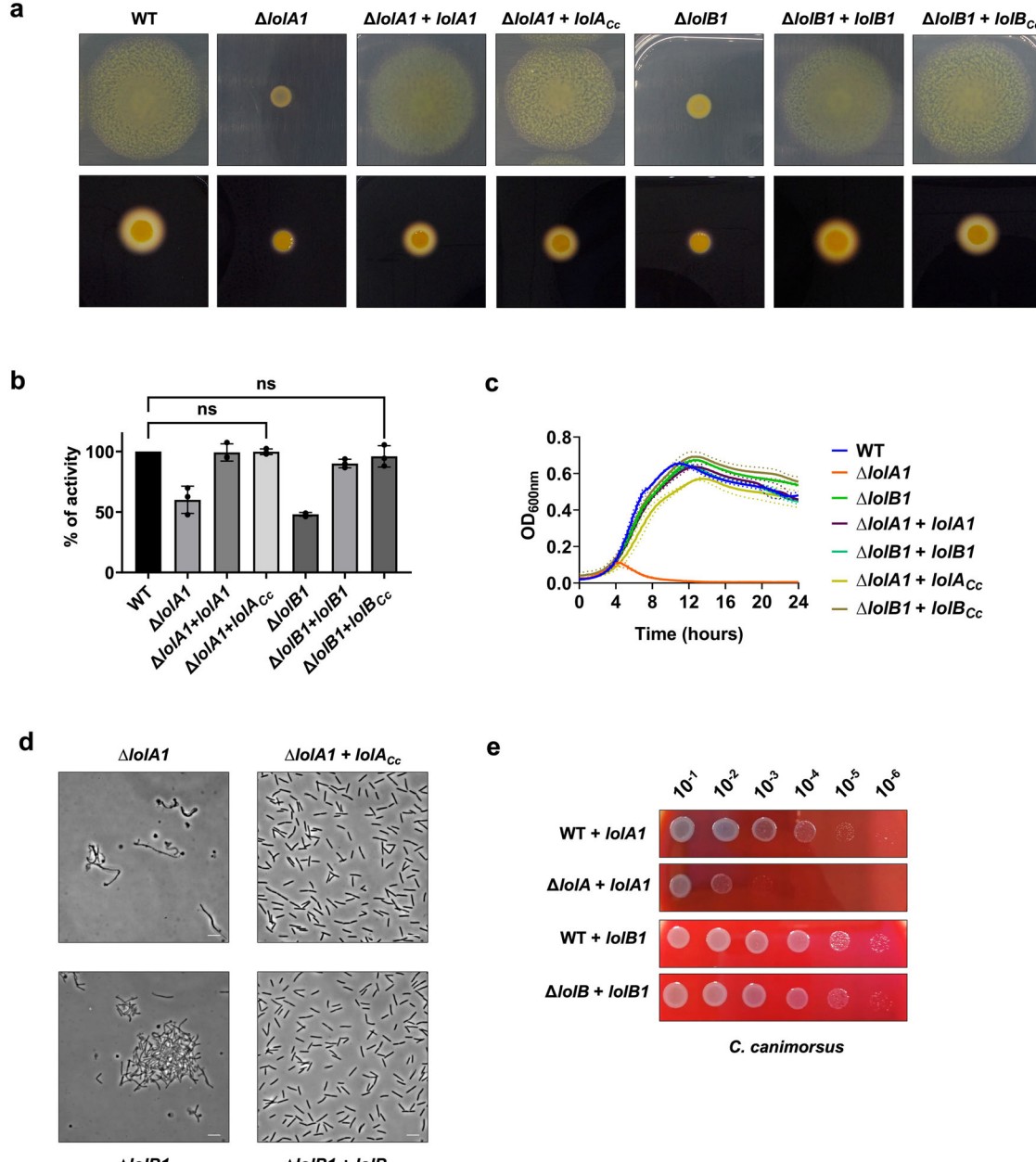

**Fig. 9 | Heterologous complementation of LolA and LolB mutants of *F. johnsoniae* and *C. canimorsus* with LolA1 and LolB1 homologs.** Gliding motility on MM agar plates after 48 h, starch degradation by T9-secreted amylases on LB starch plates after 24 h (**a**), and amylase activity of cell culture supernatants (**b**) of *lolA1* and *lolB1 F. johnsoniae* mutants complemented with *C. canimorsus lolA* (*lolA$_{Cc}$*) and *lolB* (*lolB$_{Cc}$*). Data in **b** are represented as means of $n = 3$ independent biological replicates ± standard deviation, and statistical analysis was a one-way ANOVA followed by Dunnett's multiple comparison test ($F (22, 46) = 47.47$) (ns indicates $p > 0.05$). Growth in MM liquid medium (**c**) and phase-contrast microscopy images of bacteria grown in liquid MM medium for 16 h (**d**) of *lolA1* and *lolB1 F. johnsoniae* mutants complemented with *C. canimorsus lolA* (*lolA$_{Cc}$*) and *lolB* (*lolB$_{Cc}$*) (bar = 5 μm). Data in **c** are represented as means of $n = 3$ independent biological replicates ± standard deviation. Growth of serial dilution spots of *C. canimorsus lolA* and *lolB* mutants expressing *F. johnsoniae lolA1* or *lolB1* in trans on SB plates (**e**).

In *X. campestris* and *P. aeruginosa*, it is known that *lolA* deletion and depletion, respectively, impact cell envelope stability, with increased sensitivity of the mutants to several OM stresses[39,40]. In *E. coli*, removal of *lolA* or *lolB* determines mislocalization of Lpp and RcsF that leads to cell death[41]. Here, we show that deletion of *lolA1* and, to a lesser extent, *lolB1* destabilizes the OM of *F. johnsoniae*. Hence, we observe substantial cell shape abnormalities for both mutants in the poor medium and even in the rich medium for the *lolA1* single and *lolA1lolB1* double mutants (Fig. 4b and Supplementary Fig. 3). We attempted to rationalize the weaker fitness of bacteria lacking LolA1 with our proteomic data. The severe membrane defects and cell material leakage observed for the *lolA1* mutant

(Supplementary Fig. 3) very likely resulted in the release of cytoplasmic proteins that we identified by proteomics (Supplementary Data 2). In total, the amount of four essential proteins of *F. johnsoniae* (Fjoh_0540, Fjoh_3469, Fjoh_5008, Fjoh_0105) was reduced in the OM of the *lolA1* mutant while not in the *lolB1* mutant (Supplementary Data 1). Besides, one of them is a lipoprotein homologous to *E. coli* BamD (Fjoh_3469, FC = 0.45), which is required for the activity of the BAM complex[42]. Malfunction of the BAM complex would be a sensible explanation for the great decrease in β-barrel proteins that we observed in the *lolA1* OM. 13 SusC-like proteins (out of the 21 whose amount was reduced) were decreased along with their SusD-like partner upon deletion of *lolA1*. Such a diminution of β-barrel

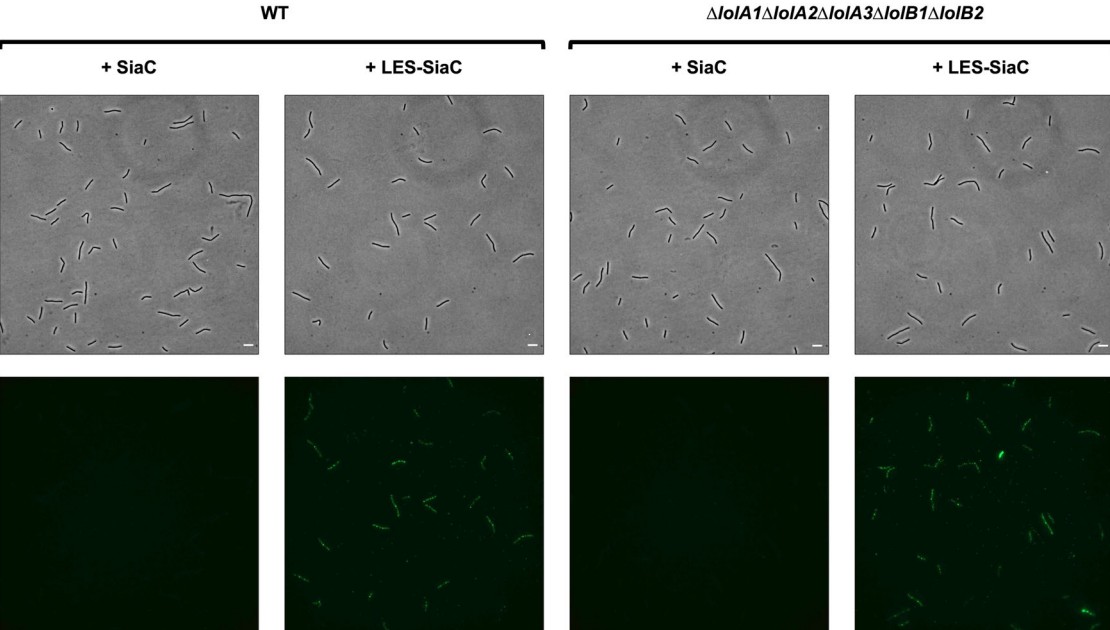

**Fig. 10 | Deletion of all LolA and LolB homologs of *F. johnsoniae* does not affect surface lipoproteins localization.** Immunofluorescence microscopy images of *F. johnsoniae* WT and *lolA1 lolA2 lolA3 lolB1 lolB2* mutant bacteria expressing sialidase (SiaC) or LES-sialidase (LES-SiaC) labeled with anti-SiaC serum (scale bar = 5 μm).

proteins is a modification of the OM proteome that might destabilize the OM, to the point of making it more sensitive to stresses (Fig. 5c). A possible explanation for the difference in fitness between *lolA1* and *lolB1* mutants could be that the IM essential complex LolCDE/F[41] becomes stalled in absence of LolA1 to pick up lipoproteins, which would obstruct or at least hinder the lipoprotein transport pathway. On the other hand, deletion of *lolB1* likely has no effect on the functioning of LolCDE/F, given their respective localization in the outer and inner membranes. The growth complementation and the partial restoration of cell shape aberrations in MM medium by MgSO₄, a known stabilizer of LPS-LPS interactions, further support the idea that deletion of *lolA1* or *lolB1* disrupts OM stability.

In this work, we explored the structure of LolB1, based on our knowledge of *E. coli* LolB. The importance of the conserved protruding loop and the leucine 74 residue for LolB1 function (Fig. 6b) strongly suggests that *E. coli* LolB and *F. johnsoniae* LolB1 work in a similar fashion. In contrast, the lack of complementation of the *F. johnsoniae lolA1* and *lolB1* mutants by *E. coli* LolA and LolB could be due to a lack of interaction as a result of differences in the structures and/or physico-chemical properties between *F. johnsoniae* LolA1 and LolB1 and *E. coli* LolA and LolB (Fig. 1, Supplementary Table 1, Supplementary Figs. 1 and 2). Our data also show that proper activity of LolB1 necessitates a disordered C-terminal domain, which appears to be a unique feature of Bacteroidota LolB1 proteins and that is absent from *E. coli* LolB.

Lastly, we found that LolA2 and LolB2 are encoded in the locus of flexirubin synthesis and transport of *F. johnsoniae*, and that these two proteins are conserved in Bacteroidota that synthesize flexirubin or flexirubin-like pigments, which are thought to be localized in the OM (Fig. 8)[19,21]. Considering the ability of LolA and LolB-like proteins to bind hydrophobic moieties, this might suggest that LolA2 and LolB2 participate in the transport of either flexirubin/flexirubin precursors, or lipoproteins encoded in the locus. To our knowledge, this would be the first example of such a specialization of LolA and LolB-like proteins to a specific pathway and we aim to test this in the next future. Concerning LolA3, we could not observe any phenotype upon its deletion but its poor conservation and low prevalence in Bacteroidota might suggest a specific role of this protein restricted to some species. Whether this protein can interact with LolB1 and/or LolB2 and participate in different pathways requires further investigation. Surprisingly, we were able to generate a viable mutant strain devoid of any

LolA or LolB protein. Essential OM lipoproteins like those part of the BAM and Lpt complexes are likely correctly localized in *F. johnsoniae* despite the absence of LolA and LoB homologs, strongly suggesting that an alternative LolA-LolB independent lipoprotein sorting pathway exists in Bacteroidota as already been suggested for *E. coli*[41]. In addition, we show that lipoproteins are still localized at the bacterial surface in the absence of LolA and LolB proteins thus suggesting that this specific class of lipoproteins does not require these elements of the Lol pathway for its localization. Overall, this study unveils the hidden diversity of LolA and LolB proteins which likely share a common function, i.e., the transport of lipidated cargos across the periplasm, but specialized over time to suit different pathways and organisms. Gene duplication events may have allowed some Bacteroidota species like *F. johnsoniae* to repurpose LolA and LolB proteins to the sole transport of lipoproteins involved in gliding motility and T9 secretion or to the synthesis/transport of outer membrane lipids such as flexirubin, for example. This diversification of lipoprotein transport routes fits with the extensive use of OM lipoproteins by Bacteroidota. We believe that these findings will contribute to a better understanding of lipoprotein trafficking in Gram-negative bacteria, a field of great importance that remains understudied outside of Proteobacteria.

## Methods
### Bacterial strains and growth conditions
Bacterial strains used in this work are listed in Supplementary Table 6. Unless otherwise stated, *F. johnsoniae ATCC 17061* UW101 strains were routinely grown in Casitone Yeast Extract (CYE) medium (10 g/l casitone, 5 g/l yeast extract, 8 mM MgSO₄ and 10 mM Tris-HCl pH 7.5) or Motility Medium (MM) (3.3 g/l casitone, 1.7 g/l yeast extract, 3.3 mM Tris-HCl pH 7.5) at 30 °C. When required, the following concentrations of antibiotics were used: Gentamicin (Gm) at 20 μg/ml, Tetracycline (Tet) at 20 μg/ml, and Erythromycin (Em) at 100 μg/ml. *E. coli* strains were grown in lysogeny broth (LB) at 37 °C and the following concentrations of antibiotics were used accordingly: Ampicillin (Amp) at 100 μg/ml, Kanamycin (Kan) at 50 μg/ml, Chloramphenicol (Cm) at 30 μg/ml and Tetracycline (Tet) at 12.5 μg/ml. *Capnocytophaga canimorsus* 5 strains were routinely grown at 37 °C in the presence of 5% CO₂ on heart infusion agar (HIA) supplemented with 5% sheep blood (SB) plates with, when required, the following concentrations of antibiotics: Gentamicin (Gm) at 20 μg/ml, Cefoxitin (Cfx) at

10 μg/ml, Erythromycin (Em) at 10 μg/ml, and Tetracycline (Tet) at 10 μg/ml.

### In silico search for LolA and LolB homologs
To search for LolA and LolB remote homologs, whole proteome structure predictions were realized for *F. johnsoniae* UW101 strain using the HHSearch suite[43] with default parameters. The structures of *E. coli* LolA and LolB were predicted using the same program and parameters. An in-house Perl script was used to compare the hit list for each of the *F. johnsoniae* proteins with the hit list for *E. coli* LolA and LolB, with a cut-off value of $10^{-3}$. Proteins with identical templates were selected as potential candidates.

### In silico modeling and characterization of LolA and LolB homologs
From the sequences of identified full-length *F. johnsoniae* LolA1 (Fjoh_2111), LolA2 (Fjoh_1085), LolB1 (Fjoh_1066), and LolB2 (Fjoh_1084), proteins were modeled using the Iterative Threading ASSEmbly Refinement (I-TASSER) protein structure predictor without specifying secondary structure preferences and additional distance restraints or templates[44]; models displaying the highest probability score were selected. Given the poor quality of the LolA3 (Fjoh_0605) I-TASSER model, the best ranking AlphaFold 3 model was chosen instead for representing the LolA3 structure[45]. The presence of signal peptides was determined using the SignalP 6.0 server[46]. As such, the N-terminal cleaved peptides were not considered. Electrostatic potentials and hydrophobicity scores were 3D mapped on the refined crystallographic structures of *E. coli* LolA (PDB entry: 1UA8) and LolB (PDB entry: 1IWL), as well as on the selected I-TASSER and AlphaFold 3 models of LolA and LolB variants from *F. johnsoniae*. For each protein, the corresponding PDB file was prepared for pKa and continuum solvation calculations with PROPKA 3.2 and PDB2PQR software in the AMBER force field, respectively. Equations of continuum electrostatics were solved using the Adaptive Poisson-Boltzmann Solver (APBS) software suite in the following conditions: pH 7.0, 298.15 K, 150 mM NaCl, 1.1 and 1.7 Å ionic radii for Na+ and Cl−, respectively. All calculation steps were carried out via the APBS/PDB2PQR website[47]. The resulting Poisson–Boltzmann electrostatic potentials were mapped on the van der Waals surface of LolA and LolB proteins, using a potential scale ranging from $-3\ k_BT/e$ to $+3\ k_BT/e$. Regarding hydrophobicity computation, residues were colored according to Eisenberg's normalized consensus hydrophobicity scale using the Color h script in the PyMOL software[48]. Scores were mapped on the van der Waals surface of LolA and LolB proteins. All the structures were visualized using the PyMOL Molecular Graphics System, Version 1.2r3pre, Schrödinger, LLC.

### Construction of *F. johnsoniae* deletion strains
Suicide plasmids (Supplementary Table 7) for deletions in *F. johnsoniae* were constructed by amplifying the 2 kb chromosomal regions upstream and downstream of the target gene with specific primers (Supplementary Table 8) and cloned sequentially (or by Gibson assembly[49] (NEB)) into the pYT354 suicide plasmid. Suicide plasmids were introduced into the appropriate *F. johnsoniae* background strain by triparental mating using *E. coli* Top10 as a donor and *E. coli* MT607 as a helper strain. Erythromycin resistance was used to select cells with chromosomally integrated plasmid. One of the resulting clones was grown overnight in CYE without antibiotics to allow for loss of the plasmid backbone and then plated on CYE agar containing 5% sucrose. Sucrose-resistant colonies were screened by PCR for the presence of the desired chromosomal modification.

### Construction of plasmids for complementation in *F. johnsoniae* and *C. canimorsus*
Genes of interest were amplified from genomic DNA using specific primers (Supplementary Table 8) with Q5 DNA Polymerase. The expression plasmids pCP23-P*ermF* or pMM47.A and the amplified genes were digested with appropriate restriction enzymes (New England Biolabs) and ligated with T4 ligase (New England Biolabs) O/N at 16 °C. The ligation products were transformed by heat shock or electroporation into *E. coli* Top10 and plated on LB Amp plates. Resistant clones were screened by PCR for the presence of the correct plasmid and constructs were checked by sequencing. Plasmids (Supplementary Table 7) were transferred to *F. johnsoniae* by triparental mating, and tetracycline resistance was used to select cells that received the plasmid. Plasmids were transferred to *C. canimorsus* by electroporation, and cefoxitin resistance was used to select cells that received the plasmid.

### Motility assays on agar plates
To assess the motility of *F. johnsoniae*, strains were grown O/N in CYE liquid medium at 30 °C. In total, $5 \times 10^7$ bacteria were collected by centrifugation for 3 min at 5000×g. The resulting pellets were resuspended in 1 ml of PBS. Three microlitres of the resuspended cells were spotted on MM plates and incubated for 48 h at 30 °C. Pictures were taken with a Nikon CoolPix L29 camera.

### Microscopic observations of cell movement on agar pads
Wild-type and mutant *F. johnsoniae* were examined for movement on agar pads by phase-contrast microscopy as described in ref. 50. Briefly, bacteria were grown overnight in MM at 25 °C without shaking, deposited on a 0.7% agar MM pad, and covered with a glass coverslip. After 5 min incubation at 25 °C, bacteria were filmed using an Axio Observer (Zeiss) microscope equipped with an Orca-Flash 4.0 camera (Hamamatsu) and the Zen Pro 3.9 software (Zeiss). Movies were mounted using the Microsoft Clipchamp version 3.1.

### Amylase activity assays
*F. johnsoniae* strains were grown O/N in CYE liquid medium at 30 °C. For plate amylase activity assays, $5 \times 10^7$ bacteria were collected by centrifugation at 5000×g for 3 min. The resulting pellets were resuspended in 1 ml of PBS. Three microlitres of the resuspended cells were spotted on LB (0.2% starch (Merck)) and were incubated for 24 h at 30 °C. A solution of 1% KI and 1% iodine was poured on the agar for starch staining, and pictures were taken with a Nikon CoolPix L29 camera. For quantitative amylase activity assays, bacterial cultures were diluted to an $OD_{600}$ of 0.05 in CYE and grown in a microplate reader in the same conditions as for the growth assays. All the strains were grown for 12 h, and after checking that they had all reached the same $OD_{600}$, cultures were retrieved and centrifuged for 3 min at 5000×g. The supernatant was collected and filtered with 0.2 μm membranes to remove bacteria. Filtered supernatant was mixed with a 1% starch solution (0.2% final) and incubated for 4 h at 30°C, 160 rpm. Finally, 143 μl of the solution was mixed with 107 μl of 3,5-dinitrosalicylic acid (DNSA) 96 mM, boiled at 99 °C for 20 min, and absorbance was read at 540 nm with a SpectraMax iD3 microplate reader (Molecular Devices).

### Membrane purification
Membranes were purified as described in ref. 26 with some modifications. *F. johnsoniae* liquid cultures (50 ml) of the WT, *lolA1* mutant, and *lolB1* mutant strains were grown O/N in CYE at 30°C, 160 rpm. In total, $2 \times 10^{10}$ bacteria were collected by centrifugation (10 min at 7000×g, 4 °C), resuspended in 4 ml of buffer (HEPES 10 mM pH 7.4, 1 mM EDTA, EDTA-free protease inhibitor cocktail from a tablet (Roche)) and lysed by 3 passages through a cell disrupter at 2.4 kbar. The lysates were then centrifuged 10 min at 2500×g, 4 °C, to pellet unbroken cells and debris. The inner and outer membranes were precipitated through a first ultracentrifugation step (107 min at 55,000 rpm with a Beckman TLA-100.3 fixed-angle rotor, 4 °C). To solubilize the inner membrane, the pellets were resuspended in a solution of HEPES 10 mM pH 7.4 and N-lauroylsarcosine sodium salt 2% (sarcosyl) (Sigma-Aldrich) and were incubated on a roller shaker for 60 min at RT. To separate the membranes, a second ultracentrifugation step (identical settings as the first) was performed, and the outer membrane pellet was resuspended in 250 μl of HEPES 10 mM pH 7.4.

## Verification of the purity of the membrane fractions

To verify for inner membrane contamination in the outer membrane fraction, an SDH activity assay was carried out[51]. During the OM purification, 200 μl were sampled right after cell lysis (total lysate fraction), 200 μl of supernatant right after the first ultracentrifugation step (periplasm + cytoplasm fraction), and 200 μl after full resuspension of the inner and outer membranes in HEPES (inner and outer membrane fraction). Inner membrane and outer membrane fractions were stored at the end of the purification. After membrane extraction, the protein concentration of the different fractions was quantified with a Bradford protein assay (Biorad). The fractions were diluted to 0.24 μg/μl of protein. A 96-well plate was filled with either 100 μl of sample or water. Then, 60 μl of the following reaction mix (final concentrations) was added to each well: 50 mM Tris-HCl (pH 8.0), 4 mM KCN, and 40 mM disodium succinate. After 5 min of incubation at RT, 20 μl of 4 mM DCIP and 20 μl of 2 mM PMS were subsequently added. Absorbance at 600 nm was measured using a SpectraMax iD3 microplate reader (Molecular Devices) every 36 s for 1 h at 25 °C. As an additional verification of outer membrane purity, 2 μg of protein was loaded on a 12% SDS-PAGE gel and visualized by silver staining. OM fractions from five independent biological replicates with low SDH activity and characteristic silver staining OM band patterns were analyzed by mass spectrometry.

## Protein digestion prior to MS

The samples were treated using Filter-Aided Sample Preparation (FASP) using the following protocol:

To first wash the filter, 100 μl of formic acid 1% was placed in each Millipore Microcon 30 MRCFOR030 Ultracel PL-30 before centrifugation at 19,980×$g$ (Ohaus 5515 R centrifuge) for 15 min. For each sample, 15 μg of proteins adjusted in 150 μl of urea buffer 8 M (urea 8 M in buffer Tris 0.1 M at pH 8.5) were placed individually in a column and centrifuged at 19,980×$g$ for 15 min. The filtrate was discarded, and the columns were washed three times by adding 200 μl of urea buffer, followed by a centrifugation at 19,980×$g$ for 15 min. For the reduction step, 100 μl of dithiothreitol (DTT) was added and mixed for 1 min at 400 rpm with a thermomixer before an incubation of 15 min at 24 °C. Samples were then centrifuged at 19,980×$g$ for 15 min, the filtrate was discarded, and the filter was washed by adding 100 μl of urea buffer before another centrifugation at 19,980×$g$ for 15 min. An alkylation step was performed by adding 100 μl of iodoacetamide ((IAA), in urea buffer) in the column and mixing at 400 rpm for 1 min in the dark before an incubation of 20 min in the dark and a centrifugation at 19,980×$g$ for 15 min. To remove the excess of IAA, 100 μl of urea buffer was added and the samples were centrifuged at 19,980×$g$ for 15 min. To quench the rest of IAA, 100 μl of DTT was placed on the column, mixed for 1 min at 400 rpm, and incubated for 15 min at 24 °C before centrifugation at 19,980×$g$ for 15 min. To remove the excess DTT, 100 μl of urea buffer was placed on the column and centrifuged at 19,980×$g$ for 15 min. The filtrate was discarded, and the column was washed three times by adding 100 μl of sodium bicarbonate buffer 50 mM (ABC), in ultrapure water) followed by centrifugation at 19,980×$g$ for 15 min. The last 100 μl were kept at the bottom of the column to avoid evaporation. The digestion process was performed by adding 80 μl of mass spectrometry-grade trypsin (1/50 in ABC buffer) in the column and mixed at 300 rpm overnight at 24 °C, in the dark. The Microcon columns were placed on LoBind tubes of 1.5 ml and centrifuged at 19,980×$g$ for 15 min. In total, 40 μl of ABC buffer was placed on the column before centrifugation at 19,980×$g$ for 15 min. Two microlitres of trifluoroacetic acid (TFA) 10% in ultrapure water were added to the filtrate (0.2% TFA final). The samples were dried in a SpeedVac, resuspended in injection solvent (acetonitrile (ACN) 2%, formic acid 0.1%) for a final concentration of 250 ng/μl, and transferred to an injection vial.

## Mass spectrometry

The digest was analyzed using nano-LC-ESI-MS/MS tims TOF Pro (Bruker, Billerica, MA, USA) coupled with a UHPLC nanoElute2 (Bruker). The different samples were analyzed with a gradient of 60 min. Peptides were separated by nanoUHPLC (nanoElute2, Bruker) on a 75 μm ID, 25 cm C18 column with integrated CaptiveSpray insert (Aurora, ionopticks, Melbourne) at a flow rate of 200 nl/min, at 50 °C. LC mobile phases A was water with 0.1% formic acid (v/v) and B ACN with formic acid 0.1% (v/v). Samples were loaded directly on the analytical column at a constant pressure of 800 bar. The digest (1 μl) was injected, and the organic content of the mobile phase was increased linearly from 2% B to 15% in 22 min, from 15% B to 35% in 38 min, from 35% B to 85% in 3 min. Data acquisition on the tims TOF Pro was performed using Hystar 6.1 and timsControl 2.0. tims TOF Pro data were acquired using 160 ms TIMS accumulation time, mobility (1/K0) range from 0.75 to 1.42 Vs/cm². Mass-spectrometric analysis was carried out using the parallel accumulation serial fragmentation (PASEF)[52] acquisition method. One MS spectra followed by six PASEF MSMS spectra per total cycle of 1.16 s.

## PEAKS proteomic analysis

Data analysis was performed using PEAKS Studio 11 with an ion mobility module and Q module for label-free quantification (Bioinformatics Solutions Inc., Waterloo, ON). Protein identifications were conducted using the PEAKS search engine with 20 ppm as parent mass error tolerance and 0.05 Da as fragment mass error tolerance. Carbamidomethylation was allowed as fixed modification, oxidation of methionine and acetylation (N-term) as variable modification. Enzyme specificity was set to trypsin, and the maximum number of missed cleavages per peptide was set to two. The peak lists were searched against the *F. johnsoniae* UW101 proteome from Uniprot (220718-5,021 entries) and a contaminants database. Peptide spectrum matches and protein identifications were normalized to less than 1.0% false discovery rate.

The label-free quantitation (LFQ) method is based on the expectation–maximization algorithm on the extracted ion chromatograms of the three most abundant unique peptides of a protein to calculate the area under the curve[53]. For the quantitation, mass error and ion mobility tolerance were set respectively to 20 ppm and 0.08 1/k0. For the label-free quantitation results, the peptide quality score was set to be ≥20, and the protein significance score threshold was set to 20. The significance score is calculated as the $-10 \log 10$ of the significance testing *p*-value (0.01). ANOVA was used as the significance testing method. Modified peptides were excluded and only proteins with at least two peptides were used for the quantitation. Total ion current was used to calculate the normalization factors.

## In silico characterization of the proteins identified by label-free mass spectrometry

The presence of signal peptide I or II was predicted using the SignalP 6.0 server[46]. Proteins devoid of any signal peptide were excluded from further analyses. Proteins with SPII (lipoproteins) were manually checked for the presence of the LES (surface lipoproteins)[16,17]. Localization of the proteins was manually checked, according to the signal peptide, the predicted 3D structure, Interpro domains, and prediction by PSORTb 3.0.3[54]. Proteins belonging to SUS-like systems were annotated with the CAZy database[55]. The essentiality of the proteins was determined thanks to a transposon sequencing (Tn_seq) analysis previously realized in our group. Proteins were functionally annotated using the MicroScope platform[56], which employs data from EggNOG 5.0[57] and EggNOG-mapper 2.1[58].

## Growth and OM stress tolerance assays

*F. johnsoniae* strains were grown O/N in CYE liquid medium at 30°C, 160 rpm. Cultures were diluted to an $OD_{600}$ of 0.05 and washed once with PBS. For growth assays, cells were resuspended in fresh CYE or MM. For stress tolerance assays, cells were resuspended in fresh CYE in the absence or presence of the following compounds: Polymyxin B (2, 20 and 40 μg/ml), EDTA (0.2 and 1 mM) and SDS (0.001 and 0.005%). Optical density at 600 nm was measured every 10 minutes during 24 h at 30°C with continuous linear shaking (567 cpm) using an EPOCH2 Microplate Reader (BioTek).

## Phase-contrast microscopy

For live imaging, cells were grown O/N in regular CYE. Cultures were then diluted to an $OD_{600}$ of 0.05 in 10 ml of CYE low magnesium (4 mM $MgSO_4$) or MM, in flasks. Cells were incubated for 16 h at 30 °C, 160 rpm, and directly spotted on a microscope glass slide. Pictures were acquired with an Axio Observer (Zeiss) microscope equipped with an Orca-Flash 4.0 camera (Hamamatsu) and the Zen Pro 3.9 software (Zeiss).

## Transmission electron microscopy

Bacterial samples for electron microscopy were prepared as described in ref. 59 with some modifications. *F. johnsoniae* was grown O/N in regular CYE. Cultures were then diluted to an $OD_{600}$ of 0.05 in 10 ml of CYE low magnesium (4 mM $MgSO_4$) in flasks and incubated for 16 h at 30 °C, 160 rpm. The fixation procedure was performed as follows: 2 ml of bacterial culture was centrifuged for 3 min at 5000×*g*, and the pellet was resuspended in 400 μl of a solution of glutaraldehyde 2.5% and cacodylate 0.1% (pH 7.4). After 2.5 h of incubation at 4 °C, bacteria were washed three times with cacodylate 0.2%, resuspended in a solution of osmic acid 2% and cacodylate 0.1%, and incubated for one additional hour at 4 °C. Then, the bacteria were washed two more times with cacodylate 0.2%. Samples were dehydrated by incubation with increasing concentrations of ethanol (30%, 50%, 70%, 85%, and 100%) at RT. For the embedding in resin, samples were first washed four times with propylene oxide and then incubated in three propylene oxide/resin solutions: 15 min in 75% propylene oxide/25% resin, 20 min in 50% propylene oxide/50% resin, and 20 min in 25% propylene oxide/75% resin. Finally, pellets were embedded in pure epoxy resin. The resin was dried by incubation at 37 °C O/N, 24 h of incubation at 45 °C, and 3 days at 60 °C. Ultrathin sections were cut with a DiATOME ultra 45° diamond knife and visualized with a FEI Tecnai T10 electron microscope at 60–80 kV. Images were acquired by the software Item of Olympus with an Olympus Megaview 1024 × 1024 pixels camera.

## Complementation of *lolB1* mutant with mutated variants of LolB1

*lolB1 (Fjoh_1066)* was amplified with oligonucleotides containing the desired mutation (L74E, Δ73–76, C17G) (Supplementary Table 8). For the deletion of the C-terminal domain (223–243), Phe223 was substituted with a stop codon. The mutated genes were then cloned into pCP23-P*ermF*, and the resulting plasmids were transferred to the *F. johnsoniae lolB1* mutant by triparental mating. To add a 6xHis tag at the C-terminus of the WT and mutated LolB1 variants, the genes were amplified with a forward oligonucleotide and a reverse one devoid of the stop codon (Supplementary Table 8) and cloned into pCP23-P*ermF* in frame with the 6XHis tag encoded in the plasmid.

## Detection of His-tagged LolB variants by Western-Blot

For Western Blot analyses, bacteria were grown in CYE liquid medium supplemented with tetracycline till the stationary phase, and $5 \times 10^8$ bacteria were collected by centrifugation at 5000×*g* for 3 min. The pellet was resuspended in 80 μl of sterile water and 20 μl of SDS-PAGE buffer 5×. After heating the samples for 5 min at 98 °C, 10 μl were loaded on a 12% SDS-PAGE gel. After migration, the proteins were transferred onto a nitrocellulose membrane with a Trans-Blot Turbo Transfer System (Bio-Rad). Proteins were detected with rabbit anti-His (1/5000) (PM032, MBL) or rabbit anti-GroEL (1/1600) (G6535, Sigma-Aldrich) primary antibodies and polyclonal swine-HRP anti-rabbit (1/5000) (P0217, Dako) secondary antibodies. Membranes were developed using a KPL LumiGLO Reserve Chemiluminescent Substrate Kit (SeraCare), and images were captured using a GE Healthcare Amersham Imager 600.

## Deletion of *lolA* and *lolB* and heterologous complementation in *E. coli*

For the deletion of *lolA* and *lolB* in *E. coli MG1655*, forward and reverse primers were designed respectively with the start and end of the gene of interest sequence flanked with regions that anneal on the pKD4 plasmid containing the Kan$^R$ gene to be amplified: oligonucleotides 8675/8679 and 8677/8680 for *lolA* and *lolB* deletion respectively (Supplementary Table 8)[60]. The *E. coli* MG1655 mini-λ-Tet strain was transformed with the pBAD33-Ara inducible plasmid expressing the following genes: *lolA, lolB, lolA1, lolB1, lolA1* and *lolB1, lolA2, lolB2, lolA2,* and *lolB2*. The plasmid-borne complemented strains were grown O/N at 30 °C in LB Tet Cm and 1 ml of the O/N liquid culture was transferred to 100 ml of LB Tet Cm.

The λ red system was then induced by placing the cultures in a 42 °C water bath for 15 min at 220 rpm. Cells were then made electro-competent and transformed with PCR fragments for the deletion of either *lolA* or *lolB* chromosomal loci by electroporation, then resuspended in 1 ml of LB 0.2% arabinose (Ara) and incubated for 1 h at 37 °C. Hundred microlitres of the cells were then plated on LB Kan Cm 0.2% Ara plates. When present, colonies grown after 24 h and/or 48 h were checked for deletion of either *lolA* or *lolB* by PCR.

## Verification of the in-trans expression of LolA and LolB homologs by mass spectrometry

The following strains were grown O/N in liquid CYE Tet for *F. johnsoniae* and LB Cm for *E. coli*: *F. johnsoniae ΔlolA1 ΔlolB1* + pCP23-P*ermF-lolA-lolB (E. coli)*, *E. coli MG1655 WT* + pBAD33− *lolA1lolB1* and *E. coli MG1655 WT*+ pBAD33-*lolA2lolB2*. Eighty microlitres of cultures at OD600 = 1 were collected and resuspended with 20 μL SDS Buffer 5×. Protein concentrations of the total lysates were determined by Pierce 660 assay (Thermo Scientific, Waltham, MA, USA) and were analyzed by Mass spectrometry. Digestion of the samples, liquid chromatography, and mass spectrometry were realized as described.

## Data processing using scaffold for protein identification

The MS/MS samples were analyzed using Mascot (Matrix Science, London, UK; version 2.7.0). For heterologous complementation samples analysis, Mascot was set up to search the FLAJ1_200130 database (5023 entries) to which the *E. coli* LolA (P61316) and LolB (P61320) protein sequences were added or *E. coli* K12-McpR_200814 database (11002 entries) to which the *F. johnsoniae* LolA1 (A5FI22) and LolB1 (A5FL25) protein sequences were added assuming the digestion enzyme trypsin. For LolB1 variants samples analysis, Mascot was set up to search the FLAJ1_200130 database (5023 entries) to which the protein sequences of *F. johnsoniae* LolB1$_{L74E}$, LolB1$_{Δ73-76}$ and LolB1$_{Δ223-243}$ all with a C-terminal 6XHis tag were added assuming the digestion enzyme trypsin. Mascot was searched with a fragment ion mass tolerance of 0050 Da and a parent ion tolerance of 15 PPM. Carbamidomethyl of cysteine and j + 59 of leucine/isoleucine indecision were specified in Mascot as fixed modifications. Deamidation of asparagine and glutamine, oxidation of methionine and acetyl of the n-terminus were specified in Mascot as variable modifications. Scaffold (version Scaffold_4.11.0, Proteome Software Inc., Portland, OR) was used to validate MS/MS-based peptide and protein identifications. Peptide identifications were accepted if they could be established at greater than 91.0% probability to achieve an FDR less than 1.0% by the Scaffold Local FDR algorithm. Protein identifications were accepted if they could be established at greater than 72.0% probability to achieve an FDR less than 1.0% and contained at least 2 identified peptides. Protein probabilities were assigned by the Protein Prophet algorithm[61]. Proteins that contained similar peptides and could not be differentiated based on MS/MS analysis alone were grouped to satisfy the principles of parsimony. Proteins sharing significant peptide evidence were grouped into clusters. Proteins of interest were checked for identification.

## Conservation of LolA and LolB in bacteroidota

A Delta-blast[62] sequence similarity search was conducted on the 30 Bacteroidota species: *Bacteroides fragilis* NCTC 9343, *Bacteroides ovatus* ATCC 8483, *Bacteroides thetaioathaomicron* DSM 2079, *Bergeyella zoohelcum* ATCC 43767, *Capnocytophaga canimorsus* Cc5, *Capnocytophaga canis* CcD38, *Capnocytophaga cynodegmi* DSM 19736, *Capnocytophaga gingivalis* ATCC 33624, *Capnocytophaga ochracea* DSM 7271, *Chitinophaga filiformis* DSM 527, *Chitinophaga pinensis* DSM 2588, *Croceibacter atlanticus* HTCC2559, *Cytophaga hutchinsonii* ATCC 33406, *Flavobacterium*

*columnare* NBRC 100251, *Flavobacterium johnsoniae* ATCC 17061, *Elizabethkingia meningoseptica* NBRC 12535, *Flavobacterium psychrophilum* DSM 3660, *Flavobacterium succinicans* LMG 10402, *Flexibacter flexilis* NBRC 15060, *Gramella forsetii* KT0803, *Kordia algicida* OT-1, *Polaribacter irgensii* 23-P, *Porphyromonas gingivalis* ATCC 33277, *Prevotella intermedia* ATCC 25611, *Prevotella melaninogenica* ATCC 25845, *Riemerella anatipestifer* ATCC 11845, *Sphingobacterium mizutaii* NBRC 14946, *Sporocytophaga myxococcoides* DSM 11118, *Xanthomarina gelatinilytica* AK20, *Zobellia galactanivorans* DsiJT, using the five LolA and LolB protein sequences of *F. johnsoniae* ATCC 17061 UW101 as queries: LolA1 (WP_081432686.1), LolA2 (WP_012023170.1), LolA3 (WP_012022695.1), LolB1 (WP_012023151.1) and LolB2 (WP_012023169.1). The genomic neighborhood of all hits with a Delta-blast E value ≤ 0.001 was inspected using MaGe[63]. The structure of the corresponding proteins was predicted using AlphaFold server (with default parameters)[45] and compared with the corresponding structures of the *F. johnsoniae* initial queries by manual inspection. This resulted in a list of 93 putative homologs. The phylogenetic tree based on NCBI taxonomy was generated via phyloT[64,65]. Conservation of the C-terminal domain of LolB1 homologs was performed by manual inspection on AlphaFold models.

### Deletion of *lolA* and *lolB* and heterologous complementation in *C. canimorsus*

To obtain the *C. canimorsus lolA* (*Ccan_16490*) mutant strain expressing *F. johnsoniae lolA1*, we transformed the pMM47-*lolA1* (Cfx[R]) plasmid into *C. canimorsus lolA* harboring pFL63 plasmid (Tet[R]) which expresses WT *C. canimorsus lolA*). We selected cefoxitin-resistant colonies and subcultured them 3 times on cefoxitin-containing SB plates. Colonies were checked for tetracycline sensitivity, and loss of the pFL63 plasmid was confirmed by PCR. To obtain the *C. canimorsus lolB* (*Ccan_17050*) mutant strain expressing *F. johnsoniae lolB1* we first transformed plasmid pMM47-*lolB1* into WT *C. canimorsus* and then deleted gene *Ccan_17050*. Deletion of *Ccan_17050* in the *C. canimorsus lolB1*-expressing strain was performed by amplification and cloning into pYT354 suicide plasmid of the 500 bases of the chromosomal regions upstream and downstream of *Ccan_17050* (Supplementary Tables 7 and 8). The suicide plasmid was introduced into the *C. canimorsus* pMM47-*lolB1* strain by triparental mating using *E. coli* Top10 as donor and *E. coli* MT607 as a helper strain. Erythromycin resistance was used to select cells with chromosomally integrated plasmid. One of the resulting clones was grown overnight on an SB plate without antibiotics to allow for loss of the plasmid backbone and then plated on an SB plate containing 3% sucrose. Sucrose-resistant colonies were screened by PCR for the presence of the *Ccan_17050* deletion.

### Immunofluorescence microscopy of sialidase

*F. johnsoniae* WT and *lolA1 lolA2 lolA3 lolB1 lolB2* strains harboring plasmid pCP23-P*ermF-siaC* or pCP23-P*ermF*-LES-*siaC* were grown O/N in CYE Tet liquid medium at 30 °C. In total, $5 \times 10^8$ bacteria were collected by centrifugation at 5000×*g* for 3 min and resuspended in 1 ml of PBS. Cells were collected by centrifugation at 4000×*g* for 5 min and resuspended in 200 μl of a PBS–BSA 1% solution and incubated for 30 min at RT. Cells were centrifuged 5 min at 4000×*g* and resuspended in 200 μl of PBS 1/2000 diluted rabbit anti-SiaC serum and incubated for 30 min at RT. Cells were washed 3 times with PBS and resuspended in 200 μl of PBS 1/500 diluted donkey anti-rabbit AlexaFluor-488TM (Thermofisher) and incubated 30 min at RT in the dark. After another 3 washes with PBS, cells were fixed with 200 μl of PFA 4% for 15 min in the dark. Cells were washed once with PBS and stored at 4 °C prior to analysis. Labeled bacteria were spotted on 1% agarose PBS pads and pictures were taken using an Axio Observer (Zeiss) microscope equipped with an Orca-Flash 4.0 camera (Hamamatsu) and the Zen Pro 3.9 software (Zeiss).

### Statistics and reproducibility

For quantitative amylase activity assays, growth curve charts and OM stress tolerance assays results are presented as means of $n = 3$ independent biological replicates with standard deviations. The statistical analyses performed are given in the figure legends. $p < 0.05$ was considered statistically significant, * $p < 0.05$, ** $p < 0.01$, *** $p < 0.001$, **** $p < 0.0001$.

### Reporting summary

Further information on research design is available in the Nature Portfolio Reporting Summary linked to this article.

## Data availability

All data generated or analyzed in this study are included in the manuscript and supplementary files. Label-free proteomics mass spectrometry data are publicly available in the ProteomeXchange Consortium via the PRIDE[66] partner repository with the dataset identifier PXD056078 and https://doi.org/10.6019/PXD056078. The mass spectrometry proteomics data for *E.coli* LolA and LolB and *F. johnsoniae* LolA1 and LolB1 heterologous expression are publicly available in the ProteomeXchange Consortium via the PRIDE[66] partner repository with the dataset identifier PXD059642, and https://doi.org/10.6019/PXD059642.The source data underlying the figures can be found in Supplementary Data 1–3 and Supplementary Information files.

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

## Acknowledgements

We are grateful to Isabelle Hamer (UNamur) for providing help with ultracentrifuges and to Paul Guiraud (UNamur) for his help with optical microscopy. We thank Gwennaëlle Louis (UNamur) for her help on amylase activity assays setup and Catherine Demazy (UNamur) for help on sample preparation for proteomics. We are grateful to the Electron microscopy facility of UNamur. We thank Régis Hallez (UNamur) and Ben Berks (University of Oxford) for kindly providing plasmids. This research was funded by the Incentive Grant for Scientific Research (MIS F.4533.20 F) from the *Fonds de la Recherche Scientifique-Fonds National de la Recherche Scientifique* (F.R.S.-FNRS, http://www.fnrs.be) to F. Renzi. T. De Smet is funded by a PhD fellowship (Aspirant) from the F.R.S.-FNRS. F. Renzi is a research associate (Chercheur Qualifié) of the F.R.S.-FNRS.

## Author contributions

T.D.S., E.B., D.D., and F.R. designed research; T.D.S., E.B., F.G., J.M., L.L., R.D., F.L., M.D., G.L.-M., C.M., D.D., and F.R. performed research; T.D.S., E.B., J.M., M.D., G.L.-M., C.M., D.D., and F.R. analyzed data; T.D.S., J.M., C.M. and F.R. wrote the paper.

## Competing interests

The authors declare no competing interests.
