## [Transparent Peer Review file · Communications Biology]

LolA and LolB are conserved in Bacteroidota and are crucial for gliding motility and Type IX secretion.

Corresponding Author: Dr Francesco Renzi

Version 0:

Reviewer comments:

Reviewer #1

(Remarks to the Author)

The Lipoprotein export pathway has been extensively studied in *Escherichia coli* and few additional bacterial models and relies on the Lol system comprising the highly conserved inner membrane ABC transporter LolCDE and periplasmic chaperone LolA. The fifth component of the Lol system, the outer membrane lipoprotein LolB, was thought to be conserved only in β and γ sub-groups of the Proteobacteria phylum. In their work Smet and co-authors, by employing an in silico prediction analysis to search for remote homologs of *E. coli* LolB, identified two likely lolB homologs in *Flavobacterium johnsoniae* a species belonging to Bacteroidetes phylum. Interestingly, *F. johnsoniae* in addition to LolB1 and LolB2 possesses also three LolA homologs (LolA1, LolA2 and LolA3). LolA1 and LolB1 seem to be present in a large number of Bacteroidetes species. The lolA2 and lolB2 homologs are part of an operon involved in the synthesis and transport of flexirubin and their presence in selected Bacteroidetes species correlate with the production of flexirubin, suggesting that the LolA2-LolB2 pair is involved in the biosynthesis of this pigment. The lolA3 homolog is present in a small subset of Bacteroidetes and its deletion does not result in obvious phenotypes. The authors focus their work on the LolA1-LolB1 pair and found that they are responsible of the sorting of a subset of lipoproteins implicated in gliding motility, in TSS9 secretion and starch degradation. Deletion of lolA1 results in more severe growth, morphology and OM permeability defects compared to lolB1 deletion suggesting a role also in maintenance of OM stability.

Despite the structural similarity of *F. johnsoniae* LolA and LolB homologs with their *E. coli* counterparts, ectopic expression of *E. coli* lolA or lolB does not complement motility defects and loss of starch degradation of *F. johnsoniae* lolA1 and lolB1 deletion mutants.

Finally, the authors show that the simultaneous deletion of the five lolA and lolB homologs in *F. johnsoniae* is not lethal, does not cause morphology defects, and does not affect the localization of surface lipoprotein suggesting that these Lol homologs are not implicated in general lipoprotein trafficking to the OM.

The identification of LolB homologs in Bacteroidetes is a significant discovery. Overall, the work is well presented, and results are convincing. Few issues that authors should address as detailed below.

Figures 2 and 4

Deletion of lolA1 and lolB1 results in remarkably similar phenotypes when gliding motility and starch degradation are assessed indicating that they are working in the same pathway. However, when assessing growth, morphology and OM permeability, lolA1 mutant displays much severe defects compared to lolB1 as if LolA1 and LolB1 are working in different pathways. Could it be that LolA1 functions as a promiscuous chaperone component of the Lol system interacting with different LolB-like OM receptors in trafficking lipoproteins?

Figure 3

The majority of proteins showing different abundance in Δ lolA1 compared to WT are predicted to be located in the cytoplasm (no signal peptide). This is likely due to the severe envelope defects displayed by the lolA1 deletion mutant which shows cell material leakage in the EM images (Figure S3). The authors should refer earlier (lines 218-219) to EM images to explain these results.

Figure 6

The lolB1 construct missing the C-terminal loop seems inefficient in complementing the gliding motility of a delta-lolB1 mutant but able to rescue starch degradation. Do the authors have an explanation of this phenotype? Also, this construct is

undetectable by western blot, can the authors comment?

Figure 7

E. coli LolA and LolB cannot complement gliding motility and starch degradation defects of the corresponding lolA1 and lolB1 *F. johnsoniae* deletion mutants. Did the authors assess whether *Ec* lolA and lolB are efficiently expressed in *F. johnsoniae*? If not, the authors should mention that lack of complementation could also be the result of insufficient level of the complementing proteins.

Figure 10

The mutants deleted for all five lol homologs do not show any defects in morphology, while the single deletion of lolA1 or the double lolA1lolB1 deletion (Fig. 4) results in severe morphology defects. Can the authors comment/discuss?

Minor points

Line 87

“...is responsible lipoprotein transport...” should be “.....is responsible for lipoprotein transport”

Line 129

Why *E. coli* LolA is underlined?

Line 145

F. johnsoniae should be written in italics

Lines 204

Based on the observation that LolA1 and LolB1 are crucial for gliding motility and T9 secretion I would suggest to change “....namely the transport of lipoprotein to the OM...” into “....namely the transport of a subset of lipoprotein to the OM...”

Lines 289-290

“Deletion of any of the other LolA and LolB proteins...” should be “.....Lack of any of the other LolA and LolB proteins”

Reviewer #3

(Remarks to the Author)

The manuscript by Tom De Smet et al mainly describes the identification and functional verification of genes involved in the Lol lipoprotein transport system in *Flavobacterium johnsoniae*, a model organism used to study gliding motility and type IX protein secretion in the phylum Bacteroidetes. The authors discovered the presence of LolB homologs in *F. johnsoniae* and other members Bacteroidetes, which was previously unrecognized. They demonstrated that *F. johnsoniae* LolA1 and LolB1 are involved in gliding motility and T9SS by impacting the transport of related lipoproteins to the outer membrane. They further showed these proteins were not interchangeable with their counterparts in *E. coli* but interchangeable with the Bacteroidetes (*Capnocytophaga canimorsus*) counterparts. Interestingly, they found that a *F. johnsoniae* mutant lacking all the five LolA&LolB proteins was still able to transport a tested lipoprotein, suggesting alternative lipoprotein transport pathways may exist in Bacteroidetes. There are a few minor issues in the manuscript to be addressed. Specific comments below:

1. I recommend changing the phylum name “Bacteroidetes” to the currently used “Bacteroidota” throughout the manuscript (see Oren, A. On validly published names, correct names, and changes in the nomenclature of phyla and genera of prokaryotes: a guide for the perplexed. *npj Biofilms Microbiomes* 10, 20 (2024). <https://doi.org/10.1038/s41522-024-00494-9>)

2. Line 121-122, 402-403, 497-503, the authors indicated several times that LolA2 and LolB2 may be involved in transport or synthesis of flexirubin. Did the authors notice any significant color/pigmentation change in the lolA2/lolB2 mutants? It seems the lolA2/lolB2 mutants look similar with the wild type in pigmentation in Figure 2A. Maybe you can try to detect the flexirubin production in the mutants by the methods used in McBride MJ, et al. Novel features of the polysaccharide-digesting gliding bacterium *Flavobacterium johnsoniae* as revealed by genome sequence analysis. *Appl Environ Microbiol.* 2009 Nov;75(21):6864-75. doi: 10.1128/AEM.01495-09. Epub 2009 Aug 28. PMID: 19717629; PMCID: PMC2772454. This experiment is not essential though.

3. Lines 172, 598, etc. It seems the authors consider non-spreading as “lack of gliding motility”. It is not always true since some bacteria or spreading mutants do have gliding motility although they do not form spreading colonies (See McBride MJ, Zhu Y. *J Bacteriol.* 2013 Jan;195(2):270-8. doi: 10.1128/JB.01962-12. Epub 2012 Nov 2. PMID: 23123910; PMCID: PMC3553832, and Nelson SS, Bollampalli S, McBride MJ. *J Bacteriol.* 2008 Apr;190(8):2851-7. doi: 10.1128/JB.01904-07. Epub 2008 Feb 15. PMID: 18281397; PMCID: PMC2293251.)

It is more accurate to use “colony spreading” instead of “gliding motility” in this manuscript in most places. For example, line 172 could be “Deletion of lolA1 and lolB1 affects colony spreading and type IX secretion”. Alternatively, the authors can determine the single cell motility of the mutants on glass or agars surfaces by phase contrast microscopy. The mutants can be described as “non-motile” or “lack of gliding motility” if they fail to move on glass or agar surfaces. None of these suggested experiments are essential though.

4. Lines 368-388. Do the authors know the expression levels of LolAB proteins in the complemented *E. coli* and *F.*

johnsoniae strains? Are they comparable with those found in the wild type strains? Does the mass spectrometry data indicate this information? What promoters were used in the heterologous expression of LolAB proteins in *E. coli* and *F. johnsoniae*?

I have the above questions because *E. coli* and *F. johnsoniae* may use very different promoters for the protein expression and wonder if the lack of complementation was because of low protein expression.

It is true that Bacteroidetes use unique promoters that are different from Proteobacteria. These Bacteroidetes promoters may function poorly in *E. coli* and vice versa (see Chen S, Bagdasarian M, Kaufman MG, Bates AK, Walker ED. Mutational analysis of the *ompA* promoter from *Flavobacterium johnsoniae*. *J Bacteriol.* 2007 Jul;189(14):5108-18. doi: 10.1128/JB.00401-07. Epub 2007 May 4. PMID: 17483221; PMCID: PMC1951883.)

5. Figures 4B, 5B, 9D, S4, it says "bright-field microscopy" in the figure legends but these images look like phase contrast microscopy, which is described in the method (line 744). Are these typos? Should they be replaced by "phase-contrast microscopy"?

6. Lines 839-847, Figure 8, and Table S6, suggest adding the strain information for each species.

Minor Revisions

7. Line 30, suggest changing "Type IX protein secretion" to "type IX secretion system", which is the full name of T9SS.

8. Line 30, suggest using the full name of *E. coli*.

9. Line 37, suggesting the abbreviation of *E. coli*.

10. Line 66, suggest moving "(T9SS)" to after "systems".

11. Line 72, suggest changing "an" to "a".

12. Lines 124, 125, 126, 129, 145, etc. suggest italicize the bacterial names throughout the manuscript.

13. Line 232, suggest adding a comma after (29, 30).

14. Lines 658, 661, etc, suggest changing rpm to x g.

15. Line 730, suggest adding a reference for the Tn_seq analysis.

16. Table S8, suggest changing "pACYC ori" to "p15A ori",

17. Table S8, suggest changing "Received from Ben Berks' lab" to "Received from Ben Berks' lab". Were the two strains from Ben Berks' lab ever published? Suggest adding refs if yes.

Version 1:

Reviewer comments:

Reviewer #1

(Remarks to the Author)

In their revised manuscript De Smet and co-authors addressed convincingly all my concerns.

I have no further questions.

Reviewer #3

(Remarks to the Author)

I appreciate the authors to perform the motility assay experiment and well address all my comments. I have no further questions.

Answer to reviewers' comments:

Reviewer #1 (Remarks to the Author):

The Lipoprotein export pathway has been extensively studied in *Escherichia coli* and few additional bacterial models and relies on the Lol system comprising the highly conserved inner membrane ABC transporter LolCDE and periplasmic chaperone LolA. The fifth component of the Lol system, the outer membrane lipoprotein LolB, was thought to be conserved only in β and γ sub-groups of the Proteobacteria phylum. In their work Smet and co-authors, by employing an in silico prediction analysis to search for remote homologs of *E. coli* LolB, identified two likely lolB homologs in *Flavobacterium johnsoniae* a species belonging to Bacteroidetes phylum. Interestingly, *F. johnsoniae* in addition to LolB1 and LolB2 possesses also three LolA homologs (LolA1, LolA2 and LolA3). LolA1 and LolB1 seem to be present in a large number of Bacteroidetes species. The lolA2 and lolB2 homologs are part of an operon involved in the synthesis and transport of flexirubin and their presence in selected Bacteroidetes species correlate with the production of flexirubin, suggesting that the LolA2-LolB2 pair is involved in the biosynthesis of this pigment. The lolA3 homolog is present in a small subset of Bacteroidetes and its deletion does not result in obvious phenotypes. The authors focus their work on the LolA1-LolB1 pair and found that they are responsible for the sorting of a subset of lipoproteins implicated in gliding motility, in TSS9 secretion and starch degradation. Deletion of lolA1 results in more severe growth, morphology and OM permeability defects compared to lolB1 deletion suggesting a role also in maintenance of OM stability.

Despite the structural similarity of *F. johnsoniae* LolA and LolB homologs with their *E. coli* counterparts, ectopic expression of *E. coli* lolA or lolB does not complement motility defects and loss of starch degradation of *F. johnsoniae* lolA1 and lolB1 deletion mutants.

Finally, the authors show that the simultaneous deletion of the five lolA and lolB homologs in *F. johnsoniae* is not lethal, does not cause morphology defects, and does not affect the localization of surface lipoprotein suggesting that these Lol homologs are not implicated in general lipoprotein trafficking to the OM.

The identification of LolB homologs in Bacteroidetes is a significant discovery. Overall, the work is well presented, and results are convincing. Few issues that authors should address as detailed below.

1) Figures 2 and 4

Deletion of lolA1 and lolB1 results in remarkably similar phenotypes when gliding motility and starch degradation are assessed indicating that they are working in the same pathway. However, when assessing growth, morphology and OM permeability, lolA1 mutant displays much more severe defects compared to lolB1 as if LolA1 and LolB1 are working in different pathways. Could it be that LolA1 functions as a promiscuous chaperone component of the Lol system interacting with different LolB-like OM receptors in trafficking lipoproteins?

R: We agree with this reviewer that LolA1 could indeed be a promiscuous chaperone for lipoproteins interacting with other yet unknown OM partners and deliver them some lipoproteins. Another possibility to explain the more severe phenotype observed in the *lolA1* mutant compared to the *lolB1* one could be that the deletion of LolA1 determines the stacking of lipoproteins in the LolCDE complex at the IM thus blocking completely the

transport of lipoproteins and determining a strong stress for the bacterium. In contrast, deletion of LolB1 would not determine such a strong effect on the LolCDE complex since even if LolA1 could not deliver lipoproteins to lolB1, LolA1 would “free” the LolCDE complex that could still work with other chaperones, maybe those of the alternative system. The fact that LolCDE are essential in *Flavobacterium* differently from LolA and LolB suggests that the LolCDE complex might have a role in delivering lipoproteins to other chaperones, maybe those of the alternative pathway. In the future, we will try to prove this hypothesis by looking at interaction partners of LolC/E by pull-down assays, hoping to identify this alternative system chaperons.

2) Figure 3

The majority of proteins showing different abundance in Δ lolA1 compared to WT are predicted to be located in the cytoplasm (no signal peptide). This is likely due to the severe envelope defects displayed by the lolA1 deletion mutant which shows cell material leakage in the EM images (Figure S3). The authors should refer earlier (lines 218-219) to EM images to explain these results.

R: We agree with this reviewer, and we modified the text and referred to EM pictures earlier in the text as suggested.

3) Figure 6

The lolB1 construct missing the C-terminal loop seems inefficient in complementing the gliding motility of a delta-lolB1 mutant but able to rescue starch degradation. Do the authors have an explanation of this phenotype? Also, this construct is undetectable by western blot, can the authors comment?

R: As the reviewer points out, the WB shows that deletion of the C-terminal domain of LolB1 (LolB1 Δ 223-243) strongly affects its expression as the protein is barely detectable. We now performed MS analysis (Table S5) to detect the expression of all LolB1 variants to compare them and to try to give an explanation to the observed phenotypes. The MS analysis shows that all LolB1 variants are expressed but with different expression levels, with the L74E variant being the most expressed one (159 spectra). Concerning the Δ 73-76 variant (82 spectra) and the LolB1 Δ 223-243 (20 spectra) this suggests that these mutations affect protein expression/ stability with a more severe effect for the Δ C-ter one. However, concerning the observed phenotypes, they cannot be ascribed to the sole different protein expression levels since the Δ 73-76 shows a good expression level (comparable to that of chromosomal LolB1 under its natural promoter in a Δ lolA1 mutant (83 spectra), Table S6) but is still unable to complement both gliding and secretion, thus suggesting that the loop deletion affects the protein function. Regarding the LolB1 Δ 223-243 mutation, despite the low protein level, this protein seems to be at least partially functional as secretion is restored while gliding only partially (Figure 6B). This mutation seems to mainly impact protein expression/stability rather than function. The lack of complete gliding restoration could be due to insufficient LolB1 protein levels that might be required for efficiently deliver gliding lipoproteins and what one could try to prove this is to express this mutant LolB1 from a stronger constitutive promoter for example.

4) Figure 7

E. coli LolA and LolB cannot complement gliding motility and starch degradation defects of the corresponding lolA1 and LolB1 *F. johnsoniae* deletion mutants. Did the authors assess

whether *Ec* lolA and lolB are efficiently expressed in *F. johnsoniae*? If not, the authors should mention that lack of complementation could also be the result of insufficient level of the complementing proteins.

R: As for LolB1 variants, we have now determined the expression level of *E. coli* LolA and LolB by MS in *F. johnsoniae* Δ lolA1, Δ lolB1 as well as in the double *lolA1lolB1* mutants (Table S6). As reported in Table S6, LolA1 and LolB1 expressed from the PermF constitutive promoter could be detected at levels comparable to that of chromosomally expressed *E. coli* LolA and LolB (Table S6). We thus believe that the complete lack of complementation observed when *E. coli* LolA and LolB are expressed in *F. johnsoniae* is not due to absence or low protein expression but rather to the incompatibility of these proteins to interact with their *F. johnsoniae* counterparts.

The same is true for *F. johnsoniae* LolA1 and LolB1 expression in *E. coli*. MS data show that these proteins are expressed in *E. coli* from the pBAD promoter to levels comparable or higher than the chromosomally expressed *E. coli* LolA and LolB (Table S6).

5) Figure 10

The mutants deleted for all five lol homologs do not show any defects in morphology, while the single deletion of lolA1 or the double lolA1lolB1deletion (Fig. 4) results in severe morphology defects. Can the authors comment/discuss?

R: The reviewer is right and looking at Figure 10 is indeed striking that the quintuple mutant bacterial cell shape looks like the wt. This is actually a result of the IF protocol. Indeed, the quintuple mutant bacteria phenotype strongly resembles that of the double *lolA1lolB1* mutant where a high number of bacteria have abnormal cell shape and form clusters when grown in CYE. Still, part of the bacterial population has a normal cell shape as for the double *lolA1lolB1* mutant (see Figure 4B). The IF protocol implies several centrifugation and resuspension washing steps of bacteria before, during and after incubation with primary and secondary antibodies, and before fixing bacteria (see methods). Multiple bacterial centrifugation and resuspension results in the loss of the abnormal cell shaped and clustered bacteria that very likely lyse and are lost during the procedure. Thus, the only bacteria which remain and can be observed at the microscope at the end are those with a normal shape (*i.e.*, more “healthy” bacteria).

Minor points

Line 87

“...is responsible lipoprotein transport...” should be “.....is responsible for lipoprotein transport”

R: Thank you, we corrected it in the revised manuscript.

Line 129

Why *E. coli* LolA is underlined?

R: It is a mistake, we corrected it in the revised manuscript, thank you.

Line 145

F. johnsoniae should be written in italics

R: Thank you, we corrected it.

Lines 204

Based on the observation that LolA1 and LolB1 are crucial for gliding motility and T9 secretion I would suggest to change “...namely the transport of lipoprotein to the OM...” into “...namely the transport of a subset of lipoprotein to the OM...”

R: We agree with the reviewer suggestion and we modified the text accordingly.

Lines 289-290

“Deletion of any of the other LolA and LolB proteins...” should be “.....Lack of any of the other LolA and LolB proteins”

R: We agree with the reviewer suggestion and we modified the text accordingly.

Reviewer #3 (Remarks to the Author):

The manuscript by Tom De Smet et al mainly describes the identification and functional verification of genes involved in the Lol lipoprotein transport system in *Flavobacterium johnsoniae*, a model organism used to study gliding motility and type IX protein secretion in the phylum Bacteroidetes. The authors discovered the presence of LolB homologs in *F. johnsoniae* and other members Bacteroidetes, which was previously unrecognized. They demonstrated that *F. johnsoniae* LolA1 and LolB1 are involved in gliding motility and T9SS by impacting the transport of related lipoproteins to the outer membrane. They further showed these proteins were not interchangeable with their counterparts in *E. coli* but interchangeable with the Bacteroidetes (*Capnocytophaga canimorsus*) counterparts. Interestingly, they found that a *F. johnsoniae* mutant lacking all the five LolA&LolB proteins was still able to transport a tested lipoprotein, suggesting alternative lipoprotein transport pathways may exist in Bacteroidetes. There are a few minor issues in the manuscript to be addressed. Specific comments below:

1. I recommend changing the phylum name “Bacteroidetes” to the currently used “Bacteroidota” throughout the manuscript (see Oren, A. On validly published names, correct names, and changes in the nomenclature of phyla and genera of prokaryotes: a guide for the perplexed. *npj Biofilms Microbiomes* 10, 20 (2024). <https://doi.org/10.1038/s41522-024-00494-9>)

R: We agree with the reviewer that the new Bacteroidetes Phylum name is Bacteroidota and we corrected it in the manuscript and supplementary data.

2. Line 121-122, 402-403, 497-503, the authors indicated several times that LolA2 and LolB2 may be involved in transport or synthesis of flexirubin. Did the authors notice any significant color/pigmentation change in the lolA2/lolB2 mutants? It seems the lolA2/lolB2 mutants look similar with the wild type in pigmentation in Figure 2A. Maybe you can try to detect the flexirubin production in the mutants by the methods used in McBride MJ, et al. Novel features of the polysaccharide-digesting gliding bacterium *Flavobacterium johnsoniae* as revealed by genome sequence analysis. *Appl Environ Microbiol.* 2009 Nov;75(21):6864-75. doi: 10.1128/AEM.01495-09. Epub 2009 Aug 28. PMID: 19717629; PMCID: PMC2772454. This experiment is not essential though.

R: As pointed by the reviewer and shown in Figure 2A and 2B, the lolA2, lolB2 and lolA2lolB2 mutants still show the characteristic “orange” coloration suggesting that these mutants still synthesise flexirubin. To see whether flexirubin is localised in the OM and if its localization is

affected in these mutants, we plan to isolate the OM as well as OMVs from the WT and from the *lolA2*, *lolB2* and *lolA2lolB2* mutants and determine if this pigment is present. This can be done by flexirubin extraction and quantification as well as by TLC. Both protocols are already established and have already been tested in our lab. The role of LolA2 and LolB2 in flexirubin transport/synthesis will be further investigated and will be the object of a further publication.

3. Lines 172, 598, etc. It seems the authors consider non-spreading as “lack of gliding motility”. It is not always true since some bacteria or spreading mutants do have gliding motility although they do not form spreading colonies (See McBride MJ, Zhu Y. J Bacteriol. 2013 Jan;195(2):270-8. doi: 10.1128/JB.01962-12. Epub 2012 Nov 2. PMID: 23123910; PMCID: PMC3553832, and Nelson SS, Bollampalli S, McBride MJ. J Bacteriol. 2008 Apr;190(8):2851-7. doi: 10.1128/JB.01904-07. Epub 2008 Feb 15. PMID: 18281397; PMCID: PMC2293251.)

It is more accurate to use “colony spreading” instead of “gliding motility” in this manuscript in most places. For example, line 172 could be “Deletion of *lolA1* and *lolB1* affects colony spreading and type IX secretion”. Alternatively, the authors can determine the single cell motility of the mutants on glass or agars surfaces by phase contrast microscopy. The mutants can be described as “non-motile” or “lack of gliding motility” if they fail to move on glass or agar surfaces. None of these suggested experiments are essential though.

R: We agree with the reviewer comment, and we determined the single cell motility of the *lolA1* and *lolB1* mutants on agar pads as suggested. The results shown in Movies S1 show that *lolA1* and *lolB1* mutants are non-motile on agar pads as a *gldJ* mutant. We thus consider these mutants as non-motile (non-gliding and non-spreading). Movie S1 has been added as supplementary movie.

4. Lines 368-388. Do the authors know the expression levels of LolAB proteins in the complemented *E. coli* and *F. johnsoniae* strains? Are they comparable with those found in the wild type strains? Does the mass spectrometry data indicate this information? What promoters were used in the heterologous expression of LolAB proteins in *E. coli* and *F. johnsoniae*?

I have the above questions because *E. coli* and *F. johnsoniae* may use very different promoters for the protein expression and wonder if the lack of complementation was because of low protein expression.

It is true that Bacteroidetes use unique promoters that are different from Proteobacteria. These Bacteroidetes promoters may function poorly in *E. coli* and vice versa (see Chen S, Bagdasarian M, Kaufman MG, Bates AK, Walker ED. Mutational analysis of the *ompA* promoter from *Flavobacterium johnsoniae*. J Bacteriol. 2007 Jul;189(14):5108-18. doi: 10.1128/JB.00401-07. Epub 2007 May 4. PMID: 17483221; PMCID: PMC1951883.)

R: This point was also raised by Reviewer 1 (point 4) and we report here the same answer we provided him/her.

We have now determined the expression level of *E. coli* LolA and LolB by MS in *F. johnsoniae* $\Delta lolA1$, $\Delta lolB1$ as well as in the double *lolA1lolB1* mutants (Table S6). As reported in Table

S6, LolA1 and LolB1 expressed from the PermF constitutive promoter could be detected at levels comparable to that of chromosomally expressed *E. coli* LolA and LolB (Table S6). We thus believe that the complete lack of complementation observed when *E. coli* LolA and LolB are expressed in *F. johnsoniae* is not due to absence or low protein expression but rather by the incompatibility of these proteins to interact with their *F. johnsoniae* counterparts. The same is true for *F. johnsoniae* LolA1 and LolB1 expression in *E. coli*. MS data show that these proteins are expressed in *E. coli* from the pBAD promoter to levels comparable or higher than the chromosomally expressed *E. coli* LolA and LolB (Table S6).

5. Figures 4B, 5B, 9D, S4, it says “bright-field microscopy” in the figure legends but these images look like phase contrast microscopy, which is described in the method (line 744). Are these typos? Should they be replaced by “phase-contrast microscopy”?

R: We apologize and indeed these are phase-contrast microscopy pictures. We corrected it in the figure legends.

6. Lines 839-847, Figure 8, and Table S6, suggest adding the strain information for each species.

R: We added the strain information in the text in lines 839-847 as well as in Figure 8 and Table S7 (formerly Table S6) as suggested by the reviewer.

Minor Revisions

7. Line 30, suggest changing “Type IX protein secretion” to “type IX secretion system”, which is the full name of T9SS.

R: Ok, we corrected it as suggested.

8. Line 30, suggest using the full name of *E. coli*.

R: Ok, we corrected it.

9. Line 37, suggesting the abbreviation of *E. coli*.

R: Ok, we corrected it.

10. Line 66, suggest moving “(T9SS)” to after “systems”.

R: Ok, we corrected it.

11. Line 72, suggest changing “an” to “a”.

R: Ok, we corrected it.

12. Lines 124, 125, 126, 129, 145, etc. suggest italicize the bacterial names throughout the manuscript.

R: Ok, we corrected it.

13. Line 232, suggest adding a comma after (29, 30).

R: Ok, we added it.

14. Lines 658, 661, etc, suggest changing rpm to x g.

R: Ok, we changed it and we also corrected the centrifuge name.

15. Line 730, suggest adding a reference for the Tn_seq analysis.

R: The Tn_seq analysis has been performed in our lab but it is not published. We thus added "Renzi lab unpublished" in the text.

16. Table S8, suggest changing "pACYC ori" to "p15A ori",

R: Ok, we changed it in Table S9 (former Table S8).

17. Table S8, suggest changing "Received form Ben Berks' lab" to "Received from Ben Berks' lab". Were the two strains from Ben Berks' lab ever published? Suggest adding refs if yes.

R: Ok, we changed it in Table S9 (former Table S8). We received these plasmids from Ben Berks' lab and we generated the strains. To our knowledge, these plasmids/strains have never been published before.